# Genetic regulation of liver lipids in a mouse model of insulin resistance and hepatic steatosis

Frode Norheim[1,2], Karthickeyan Chella Krishnan[1], Thomas Bjellaas[3], Laurent Vergnes[4], Calvin Pan[1], Brian W Parks[5], Yonghong Meng[1], Jennifer Lang[1], James A Ward[1], Karen Reue[4], Margarete Mehrabian[1], Thomas E Gundersen[3], Miklós Péterfy[1,6], Knut T Dalen[2], Christian A Drevon[2,3], Simon T Hui[1], Aldons J Lusis[1,4,*] (iD) & Marcus M Seldin[1,7,**] (iD)

## Abstract

To elucidate the contributions of specific lipid species to metabolic traits, we integrated global hepatic lipid data with other omics measures and genetic data from a cohort of about 100 diverse inbred strains of mice fed a high-fat/high-sucrose diet for 8 weeks. Association mapping, correlation, structure analyses, and network modeling revealed pathways and genes underlying these interactions. In particular, our studies lead to the identification of *Ifi203* and *Map2k6* as regulators of hepatic phosphatidylcholine homeostasis and triacylglycerol accumulation, respectively. Our analyses highlight mechanisms for how genetic variation in hepatic lipidome can be linked to physiological and molecular phenotypes, such as microbiota composition.

**Keywords** genome-wide association studies; hepatic lipidome; Hybrid Mouse Diversity Panel; non-alcoholic fatty liver disease; quantitative trait loci for lipids
**Subject Category** Metabolism
**Mol Syst Biol. (2021) 17: e9684**

## Introduction

Maintenance of hepatic lipid homeostasis is critical for many physiologic processes (Musso *et al*, 2018; Svegliati-Baroni *et al*, 2019). For example, lipid species such as ceramides and diacylglycerols appear to be key elements in non-alcoholic fatty liver disease (NAFLD), insulin resistance, and other metabolic diseases (Raichur *et al*, 2014; Ter Horst *et al*, 2017; Yang *et al*, 2018; Chaurasia *et al*, 2019). Recent advances in global lipidomics by mass spectrometry

have allowed a more comprehensive view of the hepatic lipidome (Gorden *et al*, 2015; Yang *et al*, 2018). These analyses have highlighted the complexity of lipid species and generated correlative links to several chronic diseases (Gorden *et al*, 2015; Luukkonen *et al*, 2016; Peng *et al*, 2018). Although these studies have revealed intriguing relationships between individual lipid species and metabolic traits, it has proven difficult to translate findings to a population scale using traditional approaches, such as gain- and loss-of-function studies in mice. Systems genetics provides an alternative approach for unbiased hypothesis generation based on natural genetic variation, using DNA variation as a directional anchor. This is accomplished by monitoring clinical traits and molecular information (such as gene expression or lipidomics) in a genetically diverse population and analyzing the results using genome-wide association (GWA), correlation structure, and network modeling (Civelek & Lusis, 2014).

Two recent studies have leveraged systems genetics approaches to understand how a number of hepatic lipids change across genetic backgrounds (Jha *et al*, 2018a; Parker *et al*, 2019). The first study surveyed hepatic lipids in parallel with clinical traits in a set of C57BL/6 x DBA/2J (BXD) recombinant inbred strains under two dietary conditions (Jha *et al*, 2018a). This study identified candidate genes that may modulate the abundance of a number of hepatic lipid species using GWA. They also proposed a role for cardiolipins (CL) in fatty liver progression (Jha *et al*, 2018a) and found plasma lipid signatures predicting hepatic lipid composition (Jha *et al*, 2018b). Another study utilized livers from the Hybrid Mouse Diversity Panel (HMDP) following an overnight fast. They performed liver lipidomics and proteomics and reported novel proteins regulating global lipidome structure (Parker *et al*, 2019). This study also identified plasma lipid signatures predicting hepatic triglyceride composition with several biomarkers conserved in humans. Although these studies constitute valuable resources for future studies of genetic

---

1   Division of Cardiology, Department of Medicine, University of California at Los Angeles, Los Angeles, CA, USA
2   Department of Nutrition, Institute of Basic Medical Sciences, Faculty of Medicine, University of Oslo, Oslo, Norway
3   Vitas AS, Oslo, Norway
4   Department of Human Genetics, University of California at Los Angeles, Los Angeles, CA, USA
5   Department of Nutritional Sciences, University of Wisconsin-Madison, Madison, WI, USA
6   Depatment of Basic Medical Sciences, Western University of Health Sciences, Pomona, CA, USA
7   Department of Biological Chemistry and Center for Epigenetics and Metabolism, University of California, Irvine, Irvine, CA, USA
    *Corresponding author. Tel: +1 310 825 1359; E-mail: jlusis@mednet.ucla.edu
    **Corresponding author. Tel: +1 949 824 6765; E-mail: mseldin@uci.edu

---

 

regulation of NAFLD (Seldin et al, 2019), limitations in these studies are the lack of power for association mapping (Jha et al, 2018a) and omics studies on livers after an overnight fast (Parker et al, 2019) which will likely not fully resemble lipids accumulating with NAFLD.

We now report a new resource for investigation of genetic regulation of the hepatic lipidome and its relationship to hepatic steatosis (Hui et al, 2015), insulin resistance (Parks et al, 2015), obesity (Parks et al, 2013), plasma lipids, and gut bacteria (Parks et al, 2013) in mice fed a high-fat/high-sucrose (HF/HS) diet for 8 weeks. Initially, we examined a subset of mouse strains and observed overall dietary and genetic impacts on the hepatic lipidome. Next, we performed global hepatic lipidomics on 101 HMDP strains and integrated the data with genomic variation, microbiota composition, global gene expression, and other phenotypic traits. To our knowledge, this is the most comprehensive integration of such measures in a genetically diverse population. Using association mapping, correlation, and network analyses, we identified several novel pathways regulating hepatic lipid levels and provide experimental validation to define their roles in diet-induced NAFLD and insulin resistance.

## Results

### Dietary and genetic impacts on hepatic lipidome

Initially, we evaluated the impact of a HF/HS diet on ∼ 250 lipids from the hepatic lipidome in a small group of genetically diverse mice from the HMDP. We selected three strains ($n$ = 3 mice/strain) responding differently to the HF/HS diet: the traditional C57BL/6J strain, DBA/2J, which becomes highly insulin resistant (Norheim et al, 2018) and C3H/HeJ, which carries a mutation in the Tlr4 gene regulating the lipopolysaccharide response locus (Heppner & Weiss, 1965). The hepatic lipids were measured in these strains fed a HF/HS or normal chow diet and compared using limma (Ritchie et al, 2015; Fig 1A). A large number of lipid species known to be involved in fatty liver development, such as ceramides (Chaurasia et al, 2019), were significantly changed in response to the HF/HS diet, regardless of genetic background; however, some lipids changed in a strain-specific manner, either across or between diets (Fig 1B). Particularly, the C3H/HeJ mice seemed to have a somewhat different response to a dietary perturbation for several of the phosphatidylcholine (PC) and phosphatidylethanolamine (PE) lipids than the other two strains (C57BL/6J and DBA/2J) suggesting gene-by-diet interactions. Free fatty acids (FFAs) and triacylglycerols (TAGs) with fewer carbon atoms were mostly increased after a HF/HS diet, several of the same species containing many carbon atoms decreased (Fig 1A–C). Another example showed that cholesterol esters (CE) were up- or down-regulated by HF/HS diet, dependent on the number of double bonds on their carbon backbone. Specifically, CE (C18:1) was increased and CE(C18:2) was decreased in responds to diet (Fig 1B and C). The full list of lipids impacted by diet in each strain is provided in Dataset EV1. These data indicate an interaction between genetics and diet to mediate changes in the hepatic lipidome and highlight consideration of genetic background when determining dietary effects on liver lipids.

We next expanded our survey to assay 256 hepatic lipids of 101 HMDP strains (279 mice) fed a HF/HS diet and to integrate lipidomics with other molecular layers (genome and liver transcriptome), as well as phenotypic outcomes such as HOMA-IR. We reasoned that these integrations might uncover new mechanisms by which genetic variation predisposes to metabolic alteration with involvement of liver lipids. A high degree of genetic variation was observed in the relative abundance of each lipid class compared with total lipid content (Fig 2A). For example, the most abundant lipid class (TAG) accounted from 44 to 79% of total lipids in liver and the content of PC varied > 3-fold (Fig 2A). The less abundant lipids generally exhibited greater variation across the strains. For example, ceramide-phosphatidylethanolamine (Cer-PE) and a phosphatidylinositol (PI) species varied 356-fold and 2,199-fold (Fig EV1) across the strains, respectively. Summary level statistics, such as mean abundance and variance across the 279 mice, are provided for each lipid class (Dataset EV2) and individual lipids (Dataset EV3). Not all lipid species varied substantially across strains. For example, Cer (34:2) and PC(34:1) showed minimal variation relative to the mean compared to other lipids (Dataset EV3). While analytical variation can clearly contribute to these observations, higher variation among lower abundances across genetic backgrounds has been widely appreciated for multiple omics measures and reviewed in detail (Liu et al, 2016).

### Relationships between gut microbiota and hepatic lipids

In this study, we provide several examples for how analyses can be performed on these data to infer new biologic mechanisms, where the most straightforward is correlation. While simple, analysis of correlation structure can be powerful. The intuition for assaying correlation structure is that natural genetic variation has produced a spread of complex interactions, where new relationships (either causal or reactive) can easily be inferred. For example, little is known about how individual hepatic lipid species may be affected by intestinal microbiota composition. Therefore, we performed correlation analyses to gauge genetic relationships between the hepatic lipidome and microbiota composition. Given that both of these traits appear to be highly heritable, we hypothesized that both known and new interactions could be identified (Parks et al, 2013; Org et al, 2015; Org et al, 2017). These analyses highlighted clusters of TAGs strongly correlated with the abundance of Ruminococcus, a relationship which has been observed with progression from NAFLD to non-alcoholic steatohepatitis (NASH) in humans (Boursier et al, 2016; Fig 2B). Additionally, Anaeroplasma, AF12, and Desulfovibrio showed negative correlations with many CL and lysophosphatidylcholine (LPC) species (Fig 2B). Anaeroplasma has been associated with unfavorable lipid profiles in humans (Granado-Serrano et al, 2019), but the underlying mechanisms are unclear. Desulfovibrio increases in the gut when C57BL/6J mice transition into hepatic steatosis and NASH after being treated with streptozotocin and fed a high-fat diet (Xie et al, 2016). To our knowledge, no previous study has observed an association between Anaeroplasma, AF12, and NAFLD. Our analyses suggest that the gut levels of Anaeroplasma, AF12, and Desulfovibrio might affect the hepatic levels of several hepatic lipids such as CL and LPC; however, these relationships require direct experimentation to prove directionality and causality. Because many lipids were strongly intercorrelated, we next

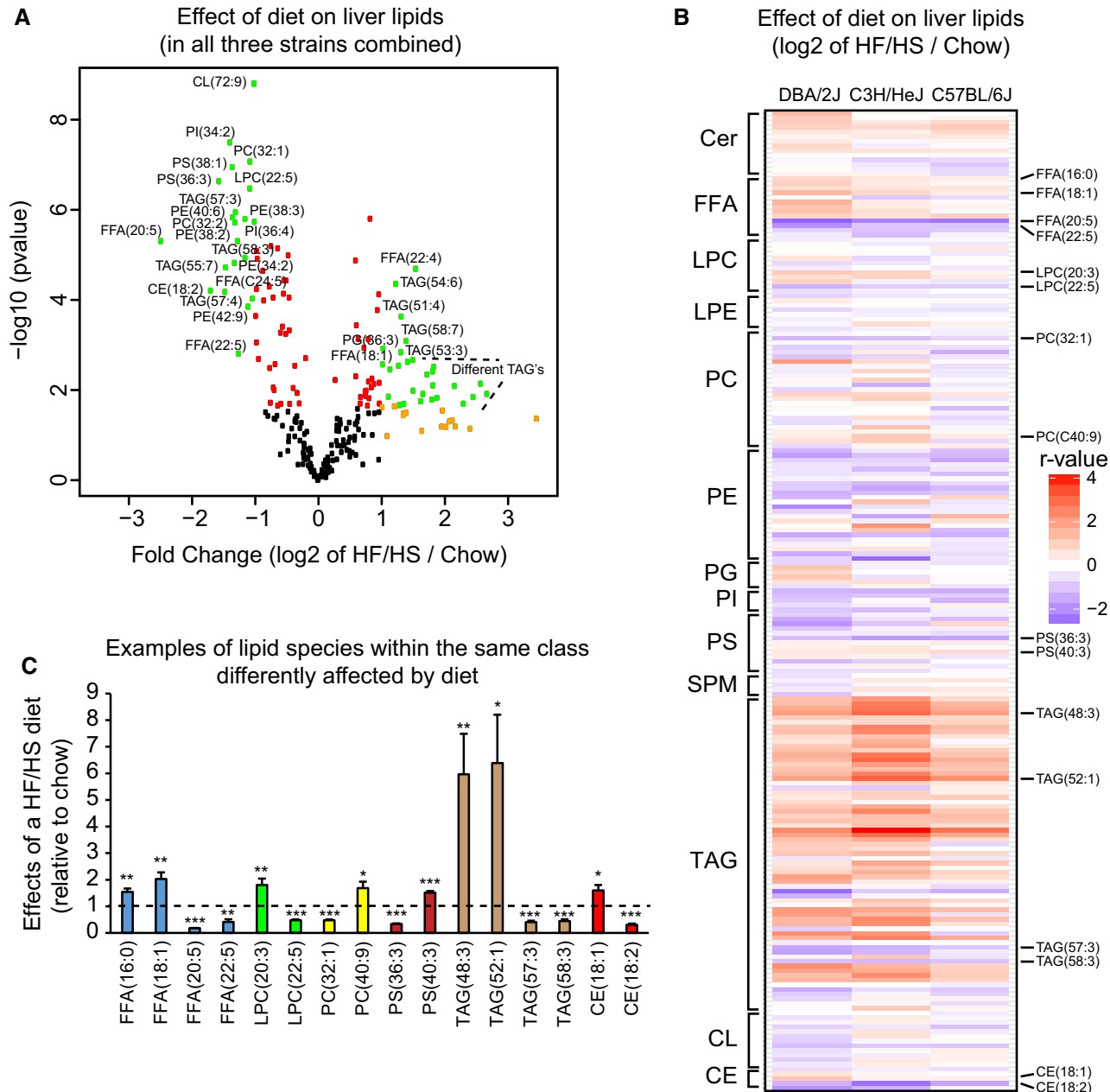

**Figure 1.   Dietary and genetic effects on the hepatic lipidome.**

A   Volcano plot of the fold change (*x*-axis) plotted against significance (*y*-axis) of lipids changing upon HF/HS feeding. Lipids are colored according to fold change (log₂, absolute) > 1 (orange), *P*-value < 0.05 (red), or both (green). *P*-values calculated from differential expression using limma.

B   Heatmap of the fold change (log₂) of each lipid in HF/HS compared to chow diet. Only lipid species detected in all mice are shown.

C   Examples of different hepatic lipids within one class that are regulated in different directions in HF/HS fed as compared chow fed mice.

Data information: Cer, ceramide; FFA, free fatty acids; LPC, lysophosphatidylcholines; LPE, lysophosphatidylethanolamines; PC, phosphatidylcholines; PE, phosphatidylethanolamines; PG, phosphatidylglycerols; PI, phosphatidylinositols; PS, phosphatidylserines; SPM; sphingomyelins; TAG, triacylglycerols; CL, cardiolipins; CE, cholesterol esters. Specific comparison results are provided in Dataset EV1. *N* = 3 male mice per strain and diet group. Data represent mean ± SEM. **P* < 0.05, ***P* < 0.01, ****P* < 0.001. *P*-values calculated using a Student *t*-test (two-tail) compared with chow group.

aggregated lipid species into modules of correlated members using weighted gene co-expression network analysis (WGCNA) (Lang-felder & Horvath, 2008). Lipid species clustered into 12 discrete modules, some were predominantly a single class, whereas others included lipids from multiple classes (Figs EV2 and EV3, Dataset EV4). For example, a majority of the TAGs (36/47) and PCs (9/22) clustered into single modules (turquoise and magenta, respectively). Module membership for every lipid from this analysis is provided in

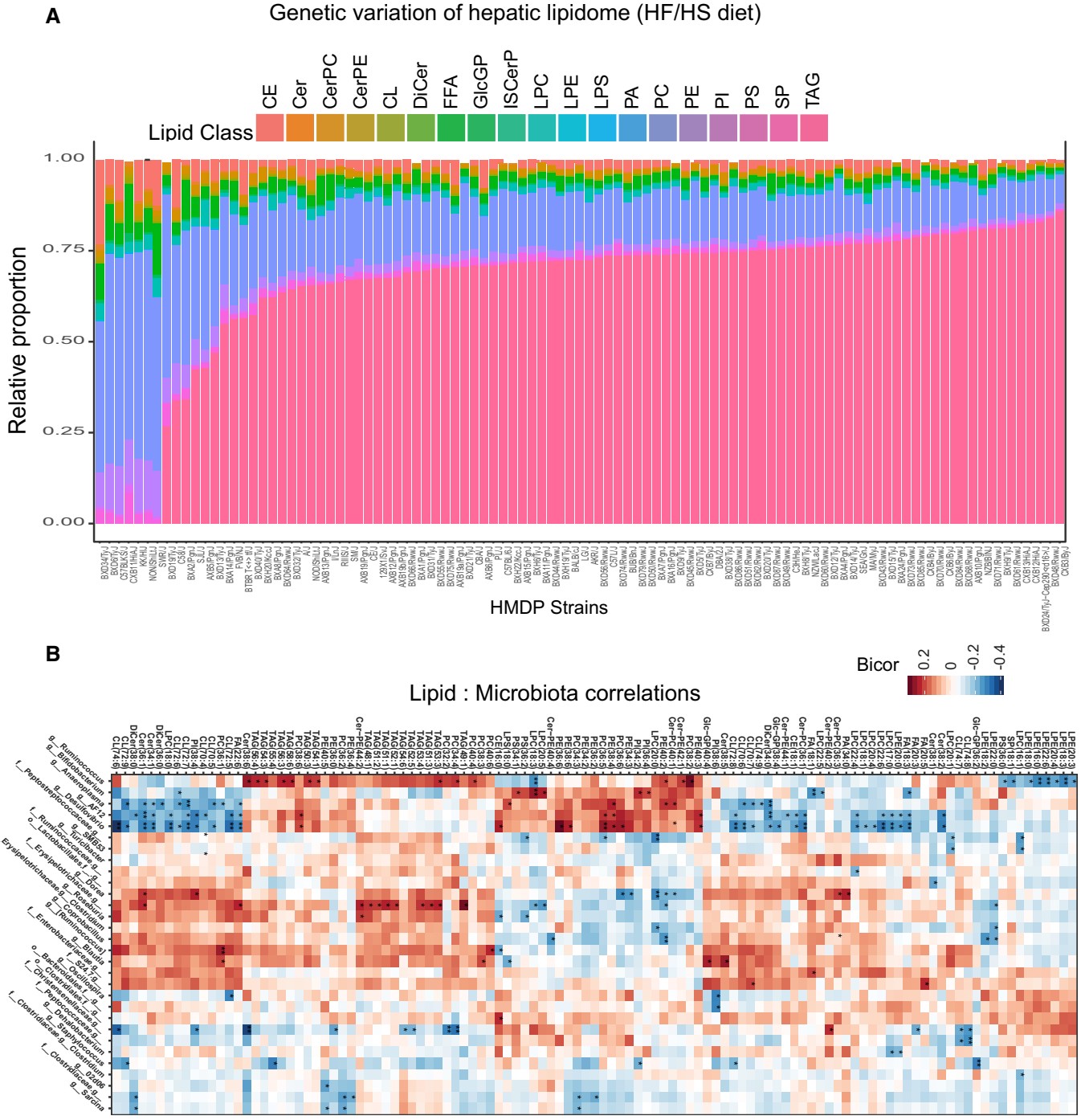

**Figure 2. Genetic variation of hepatic lipidome in the HMDP.**

A  The relative genetic variation of hepatic lipidome composition; all lipids were quantified in proportion to the total lipidome. Each lipid class is shown in a different color where differences can be observed across the strains.

B  Heatmap showing correlations between different lipid species (x-axis) and the abundance of gut microbes (y-axis). Microbes were summarized at the levels of order (o_), genus (g_), or family (f_). *$P < 0.05$, **$P < 0.01$ P-values were calculated based on significance of regression (students test) and adjusted for multiple comparisons (FDR = 0.05).

Dataset EV4. We also assessed relationships between microbiome abundance profiles and these lipid modules (Fig EV2). This approach highlighted how intercorrelated lipid groups could better inform relationships with gut bacteria. For example, several lesser-abundant species such as *Adlercreutzia* and *Desulfovibrio* showed modest correlation with individual lipids species but were strongly

correlated with a specific module (red, Fig EV2), which was composed exclusively of FFAs. While these genera have been observed to change in the context of inflammatory bowel disease (Bajer *et al*, 2017), little is known about their functional roles.

## Coregulated lipids are strongly correlated with phenotypic traits

We next focused our WGCNA analysis of specific coregulated lipid modules on their relationships with clinical traits. As suggested above, lipids of the same class were generally correlated with each other across the HMDP strains (Fig 3A). This is consistent with previous observations and was especially apparent for TAGs (Jha *et al*, 2018a). There were also several examples of strong correlations between lipid classes, such as phosphatidylserines (PS) correlating with phosphatidylinositols (PI), as well as lysophosphatidylethanolamine (LPE) and LPC showing strong correlations with FFAs (Fig 3A). Because analysis of correlation structure between lipids is a key component of several analyses, we have provided the midweight bicorrelation coefficient and corresponding *P*-value for all lipid pairs in Dataset EV5. To examine further the relationships being driven by genetic architecture, we selected several relevant phenotypic traits and integrated these with separate lipid species (Fig 3B). Several key lipids showed strong correlation with traits consistent with previous studies. As examples, the levels of some hepatic ceramides and PEs correlated negatively with plasma glucose levels and body fat percentage, respectively (Fig 3B). These data show that genetic variation may drive hepatic lipids to cluster within or between classes and that pairwise relationships exist between individual lipid species and phenotypic traits.

To obtain a comprehensive picture of how lipid subgroups may relate to these traits, we adopted two network-based approaches. First, a correlation-based network map was constructed, where connections between components can be visualized through strength of correlation (Fig EV4). This cumulative network showed that metabolic syndrome traits such as body weight and HOMA-IR were strongly correlated with several lipid species like Cer-PE lipids. In contrast, plasma glucose concentration was more strongly correlated with several PC species (Fig EV4). Next, we asked if lipid modules identified from WGCNA (above) were correlated with the same traits. The turquoise and magenta modules both showed strong positive correlations with body weight and plasma insulin concentration (Fig 3C). All the CLs (13/13) clustered into a single module (blue) which showed a negative association with liver cholesterol, and plasma HDL, TAG, and glucose (Fig 3C). Other modules were more diverse in their membership, but still showed strong correlations with phenotypic traits. For example, the purple module contained lipid species from seven different classes (Fig 3C). When combined, this module showed significant correlations with body weight as well as plasma insulin and HDL (Fig 3C). Taken together, these data show that within a broad network, close connections can be observed between specific lipid species, global lipid classes, and traits.

## Association mapping prioritizes high-confidence genes involved in hepatic lipid metabolism

Genetic loci controlling lipid levels were first identified using GWA, and the genes present in the loci were further examined for evidence of genetic variation in gene expression. We have previously determined a genome-wide significance threshold of $P = 4.1 \times 10^{-5}$ for the HMDP (Bennett *et al*, 2010). Using this threshold, we identified 407 quantitative loci for 140 lipid species (Dataset EV6). Associations between genetic markers and gene expression levels were performed, and local expression quantitative trait loci (local eQTL), presumably acting in *cis*, were identified. Gene expression can be controlled by a combination of both cis- and trans-acting elements. Genes whose *cis* components of gene expression were correlated with lipid levels were considered strong causal candidates (Dataset EV7). For example, a locus for several hepatic LPCs (Datasets EV6 and EV7), with a peak SNP rs27364570 (Fig 4A), was also associated with the cis component of the expression of *Pex16* (Fig 4B), encoding peroxisomal biogenesis factor 16. *Pex16* expression was also correlated with LPC levels and certain clinical traits, including fat mass and liver mass (Fig 4C). Mediation analysis supported a causal role for *Pex16* (Fig EV5). Several lipid loci also harbored genes previously known to be involved in lipid metabolism. For example, a number of genes involved in NAFLD-related traits like estrogen-related receptor alpha (*Esrra*) (B'Chir *et al*, 2018), reticulon 3 (*Rtn3*) (Xiang *et al*, 2018), and proprotein convertase subtilisin/kexin type 5 (*Pcsk5*) (Iatan *et al*, 2009) were all located within loci for various liver TAGs and exhibited a local eQTL where the cis component of the expression correlated with the lipid (Dataset EV7). In total, we identified 76 loci whose *cis* component of gene expression was correlated with lipid levels (55 unique lipid species) as listed in Dataset EV7. Below we validate two novel regulators of lipid levels and metabolic traits.

## Role of *Map2k6* in the control of hepatic TAG(C48:2) and response to a HF/HS diet

TAGs were the most abundant hepatic lipids (Fig 2B) and showed strong correlations with metabolic traits (Fig 3B and C). Given the clear role of TAG accumulation in hepatic steatosis, we searched for genomic regions which associated with multiple TAG species. We observed that TAG(56:3), TAG(54:4), TAG(48:2), TAG(48:1), and TAG(48:0) all mapped to approximately the same area on chromosome 11 (Fig 5A, Dataset EV2). This locus included only three potential candidate genes: ATP-binding cassette subfamily A member 5 (*Abca5*), ATP-binding cassette subfamily A member 6 (*Abca6*), and mitogen-activated protein kinase 6 (*Map2k6*). Integration with hepatic gene expression revealed that *Map2k6* was regulated in *cis* by the same loci (Fig B). The hepatic levels of TAG (C48:2) also showed a significant association with the *cis* component of *Map2k6* expression (Dataset EV4). Further, the expression of *Map2k6* correlated significantly with a majority of TAGs, in addition to TAG(C48:2) (Fig 5C). There were, unfortunately, no probes for *Abca5* and *Abca6* on our microarray platform. To test the hypothesis that genetic variation in the *Map2k6* gene was causal for accumulation of hepatic TAG, male C57BL/6J mice were administered $1 \times 10^{12}$ PFU/mouse adeno-associated virus (AAV) expressing either GFP or *Map2k6* cDNAs under a thyroid-binding globulin (TBG) promoter and subsequently fed a HF/HS diet for 8 weeks (Fig 5D). Viral administration led to a substantial increase in liver MAP2K6 protein levels compared with the GFP control (Fig 5E). The hepatic lipids were quantified, revealing a significant reduction in total TAG in the AAV-*Map2k6* group (Fig 5F). Moreover, hepatic

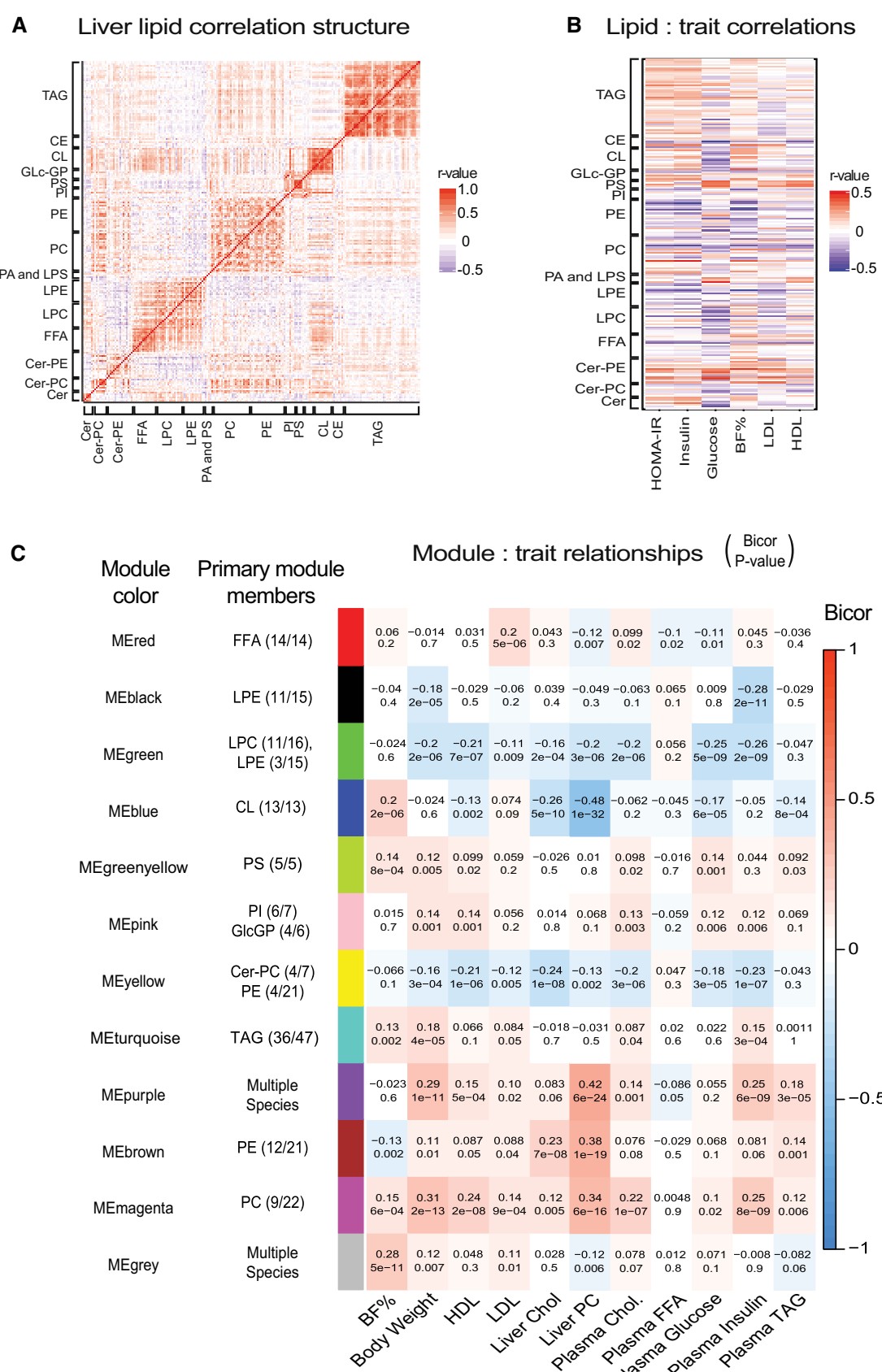

**Figure 3.**

**Figure 3.  Genetic lipidome structure and correlation with phenotypic traits.**

A  Heatmap showing correlations among hepatic lipids.

B  Heatmap showing concordance between different lipid species (class listed on *y*-axis) and certain phenotypic traits on the *x*-axis.

C  Results from WGCNA analyses, where lipids were separated in 12 modules and, labeled distinct colors, based on their internal correlations. Primary lipid classes, which comprise each module, are listed as primary module members, with the number of species in each module/total number of species detected. The correlations between each of these lipid modules and relevant phenotypic traits are shown as a heatmap, where bicor (top) and *P*-value (bottom) are listed. *P*-values were calculated based on significance of regression (students test) and adjusted for multiple comparisons (FDR = 0.05).

PC levels showed modest, but significant reductions in the *Map2k6* group (Fig 5G). This novel regulatory mechanism affecting hepatic TAGs also appeared to be relevant for other physiologic outcomes. Overexpression of *Map2k6* significantly blunted the increase in body fat percentage typically associated with a HF/HS diet (Fig 5H), as well as reduced upregulation of the plasma concentrations of glucose (Fig 5I) and insulin (Fig 5J).

## Interferon-activable protein 203 (*Ifi203*) influences hepatic PC (C38:3) levels

We identified a locus (peak SNP at rs31614030) significantly associated with the expression of a proximal gene, interferon-activable protein 203 (*Ifi203*), hepatic PC(C38:3) levels, and plasma insulin concentrations (Fig 6A–D). In addition, a strong correlation was observed between the PC(C38:3) levels, *Ifi203* expression, and insulin concentration (Fig 6E–G). While other genes (including interferon-activable family members) within the same locus showed strong associations with the peak SNP, albeit not as significant, *Ifi203* was the only one which also correlated in directions consistent with the genetic effects. The surrounding genome view of the locus and *P*-value of all genes detected on our arrays are provided in Fig EV6. Next, we examined the effect of *Ifi203* knockdown on hepatic lipid levels and plasma insulin *in vivo* (Fig 6H). Mice were fed a HF/HS diet for 4 weeks to induce hepatic steatosis, then administered an adenovirus ($2 \times 10^9$ PFU/mouse) containing either a scrambled control or a shRNA targeting *Ifi203* expression under a ubiquitous CMV-U6 promoter (Su *et al*, 2008). The vector containing sh-*Ifi203* resulted in a ~ 60% reduction in *Ifi203* mRNA expression (Fig 6I) and a significant increase in total hepatic PC levels (Fig 6J). To investigate potential links between *Ifi203* and PC concentrations, we monitored gene expression of enzymes involved in synthesis or catabolism of PC in livers of the same mice. We observed a significant increase in mRNA expression of liver phosphatidylethanolamine N-methyltransferase (*Pemt*) when *Ifi203* was knocked down (Fig 6K). Given that the primary function of *Pemt* is to catalyze conversion of PE to PC by sequential methylation in the liver, this seems a plausible mechanism for regulating total PC levels. Although not statistically significant (possibly due to the limited time of adenoviral expression or degree of knockdown), mice receiving the sh-*Ifi203* virus trended toward higher levels of total hepatic TAG levels (Fig 6L) and plasma insulin concentration (Fig 6M).

## Discussion

We report an integrative genetics analysis of 256 lipids from the hepatic lipidome across 101 diverse inbred strains of mice fed a HF/

HS diet. Our analyses included lipid interactions with diets, with disease traits such as insulin resistance and obesity, with global gene expression in liver and adipose, and with the gut microbiome. We were able to identify quantitative trait loci for about 60% of the lipid species using a stringent threshold for genome-wide association. Based on association mapping and modeling of gene expression data, we identified *Ifi203* and *Map2k6* as novel lipid metabolism regulators. We also carried out analyses relating to gut–microbiome–lipid interactions that confirmed several previously established relationships and highlighted potentially novel connections. Our results provide a rich resource for future experimental studies of lipid metabolic regulation and the relationship of hepatic lipids to diet-induced disease traits.

To dissect the interactions between hepatic lipids and traits in the HMDP, we performed several different analyses. Initially, we surveyed global correlation structure and observed many previously described interconnections between lipids and clinical traits. For example, genetic diversity causing variation in several Cer-PE species was linked to traits such as body weight and HOMA-IR. This is in accordance with previous studies in two different transgenic mouse models where it has been shown that a reduction in plasma membrane sphingomyelins improves insulin sensitivity and ameliorates high-fat induced obesity (Li *et al*, 2011). Direct genetic modulation of enzymes affecting the ceramide pathways in mice like dihydroceramide desaturase 1 may drive insulin resistance and hepatic steatosis (Chaurasia *et al*, 2019). Our overall network view of lipids and traits allowed us to visually evaluate relationships, where we found that increased levels of certain sphingomyelins correlate negatively with plasma insulin within an interconnected network with ceramides. The relationship between hepatic sphingomyelin and ceramide levels has been established in mouse models (Kusminski & Scherer, 2019), but our data additionally suggest these connections are specifically relevant for regulation of plasma insulin.

To dissect causal genetic interactions, we then performed GWA. Because human GWAS have limited ability to access tissues and control for the environment, genetic reference panels in model organisms such as Drosophila and mice have become attractive alternatives to complement human studies (Churchill *et al*, 2004; Bennett *et al*, 2010; Mackay *et al*, 2012; Jha *et al*, 2018a). One advantage of the HMDP is that it allows genetic power and resolution, which may reduce number of candidate genes as compared to alternative approaches (Seldin *et al*, 2019). HMDP has been utilized to study hepatic lipids in chow fed animals after an overnight, prolonged fast (Parker *et al*, 2019). In our present study, we fed the mice a HF/HS diet to investigate the hepatic lipidome after diet-induced hepatic steatosis. We excluded lipids not identified in more than 50% of the strains to limit false-positive associations. It is worth noting that we used the same GWA significance threshold as previous HMDP studies mentioned above. While this threshold

**A**

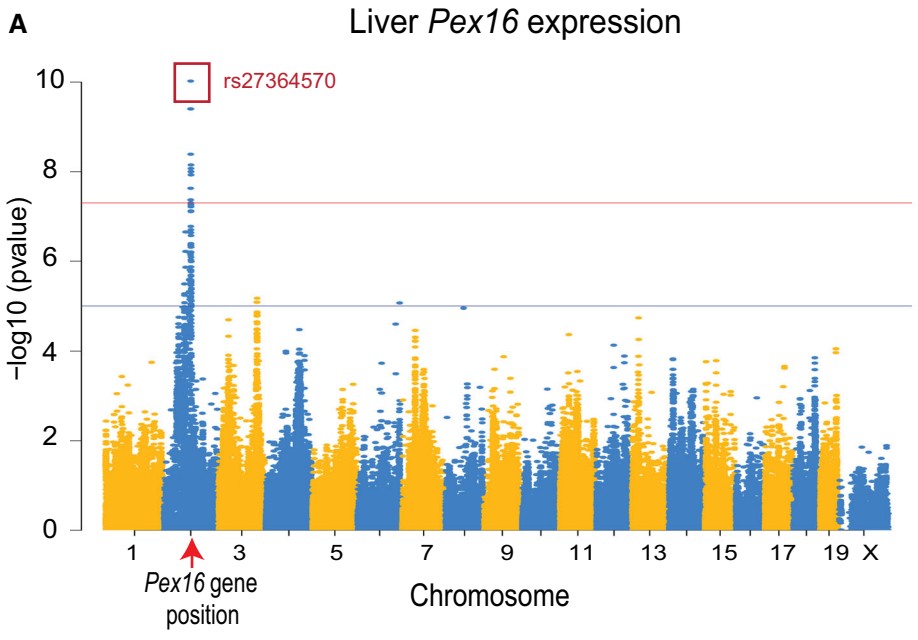

**B**

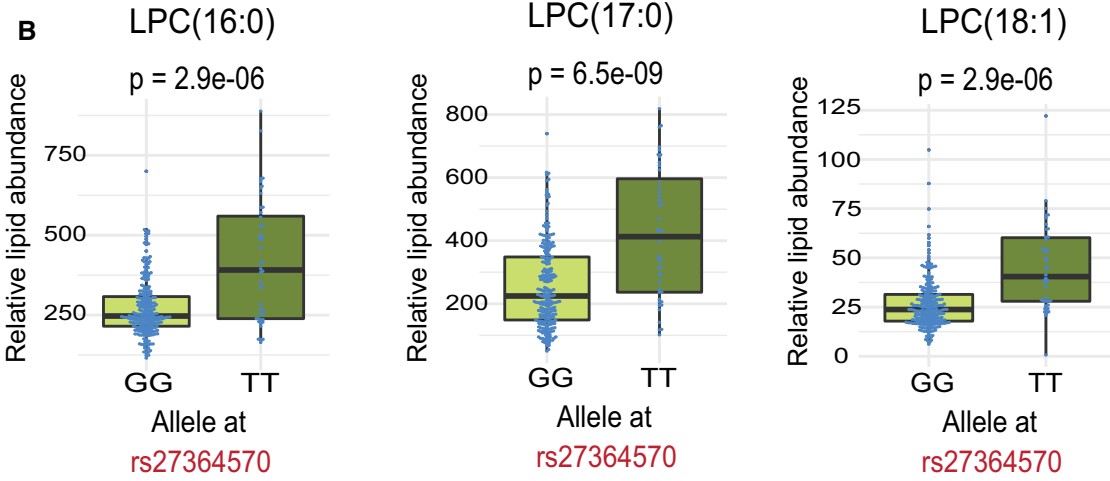

**C**

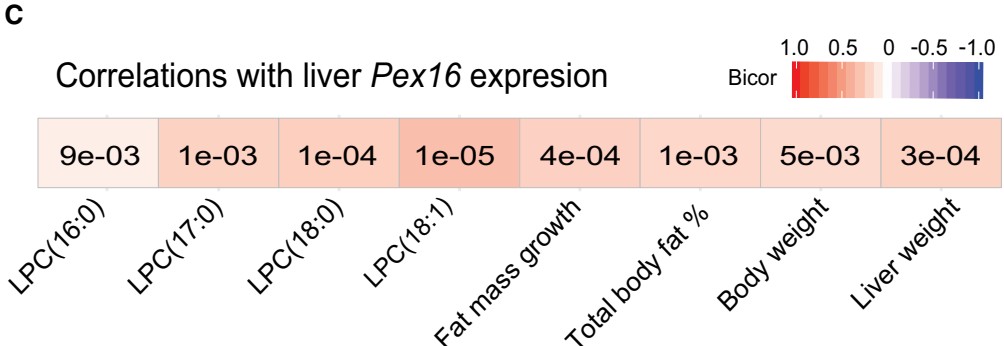

**Figure 4.**

**Figure 4.  Liver Pex16 is a novel regulator of hepatic LPC.**

A   Manhattan plot of genome-wide association for the levels of hepatic Pex16 transcript, where the only significant locus appears directly surrounding the genomic location (red arrow). Significant cutoffs are shown for FDR (blue) and Bonferroni (red). The peak SNP (rs27364570) is highlighted with a dark red box. *Y*-axis shows the −log$_{10}$ (*P*-value) vs. *x*-axis showing each SNP measured. *P*-values for GWAS associations were calculated using FaST-LMM.

B   Allelic distribution comparing GG vs. TT (*x*-axis) for the peak SNP of the Pex16 association (rs27364570), where the abundance of each LPC species (*y*-axis) showed significantly different levels depending on the allele. *P*-values for GWAS associations were calculated using FaST-LMM. Boxplots show mean (middle line), 25–75% quantiles (colored box), and 5–95% quantiles (vertical lines).

C   Correlations between expression of hepatic Pex16 with LPC species and phenotypic traits. Box color indicates bicor value, where all relationships are positive and number in each box shows *P*-value for each correlation. *P*-values were calculated based on significance of regression (students test) and adjusted for multiple comparisons (FDR = 0.05).

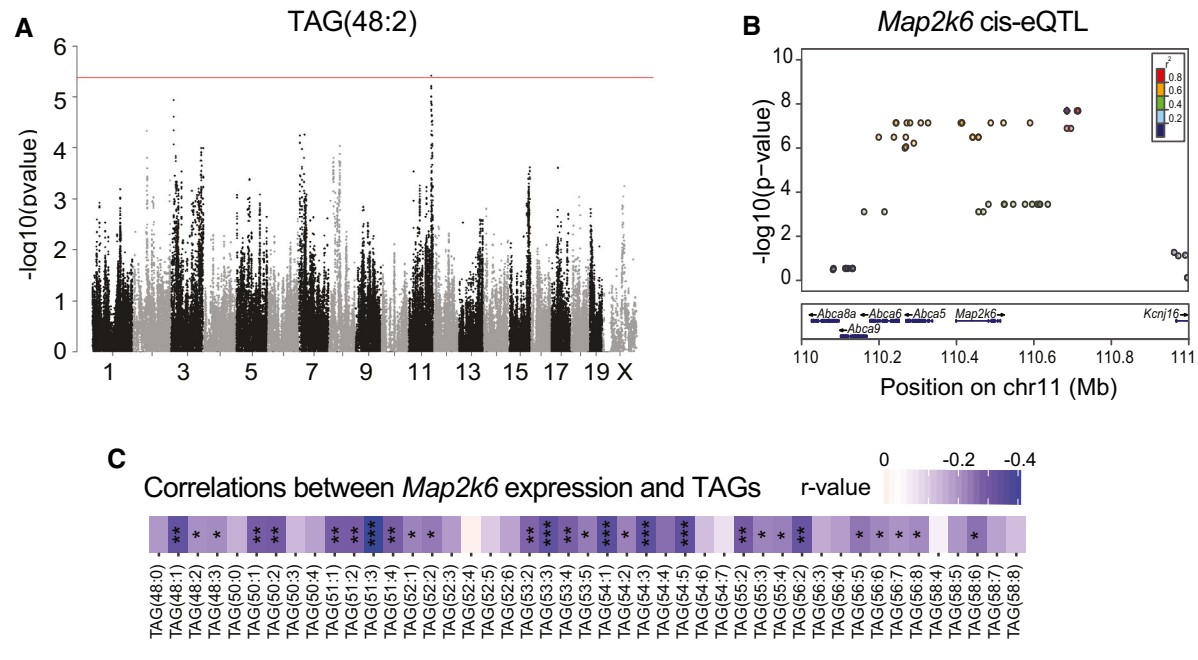

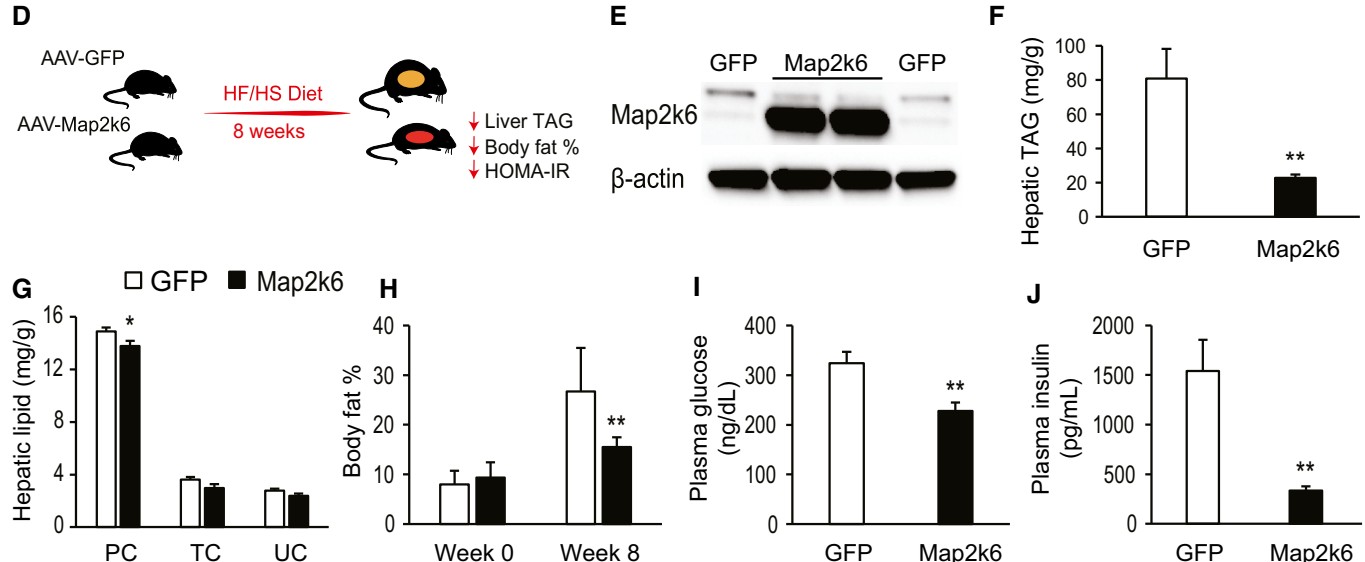

Figure 5.

**Figure 5. Map2k6 regulates hepatic TAG levels and improves metabolic profile.**

A  Manhattan plot of genome-wide association for TAG(48:2). Red line shows Bonferroni-corrected threshold of significance calculated based on FaST-LMM $P$-values.
B  LocusZoom plots showing the focused genomic region ($x$-axis) plotted against the $-\log_{10}$ ($P$-value) of association for liver mRNA expression of Map2k6. $P$-values for GWAS associations were calculated using FaST-LMM.
C  Correlations between hepatic expression of Map2k6 and all TAGs identified in the study. Blue represents negative correlations.
D  Experimental design for validation of Map2k6 as a regulator of hepatic TAGs and phenotypic traits, which changed as a result.
E  Western blots of liver homogenate using anti-Map2k6 and anti-β-actin.
F  Comparison of total hepatic TAG between Map2k6-overexpressing mice (black bar) and control mice (empty bar).
G  Differences in total phospholipid (PC), total cholesterol (TC), and unesterified cholesterol (UC) between Map2k6-overexpressing mice (black bars) and control mice (empty bars).
H  Body fat % of both experimental cohorts prior to (week 0) or 8 weeks on a HF/HS diet.
I, J  Plasma concentration of glucose (I) and insulin (J) at the end of the 8 weeks of study.

Data information: *$P < 0.05$, **$P < 0.01$, ***$P < 0.001$ calculated for the significance of correlation (C) or a Student $t$-test between groups (F–J). Data represent means ± SEM ($n$ = 10 control mice and 7 Map2k6-overexpressing mice). $P$-values were calculated based on significance of regression (students test) and adjusted for multiple comparisons (FDR = 0.05).

has been robust across HMDP studies, there are many considerations for interpreting GWAS results. For example, distribution of traits, population structure, and allele frequencies within a population can influence results of GWAS. Therefore, it is key to integrate GWAS results with other analyses (e.g., GWAS of multiple biological layers or correlation structure) and experimentation to gain confidence in underlying mechanisms. This allowed investigation of GWA of 220 separate hepatic lipids in mice with disrupted metabolic homeostasis, where about 60% were significantly associated with genomic loci. We note that about 65% of these lipid QTLs mapped to more than one locus, indicating polygenic regulation. This is comparable to a previous study on hepatic lipids in the BXD mouse genetic population showing polygenic regulation for about 50% of the lipids (Jha *et al*, 2018a). Like other complex traits, hepatic lipids are likely to be regulated by many genes, where changes in one lipid class/species will in most cases also influence levels of multiple others in the same pathway. For these reasons, it is key to integrate multiple types of analyses when analyzing system genetics resources.

Our results provide the basis of a systems genetics resource for integrating genetic regulation of hepatic lipids with hepatic lipid levels. To validate our resource, we selected two identified high-confidence candidate genes and provided preliminary evidence that genetic variation in the *Ifi203* and *Map2k6* genes alters liver PC and TAG concentrations, respectively. In selecting candidate genes to test, there are several important considerations. For example, most genes in linkage disequilibrium will be correlated with each other, making it difficult to infer a single causal candidate. Causal inference tests, such as cis-expression correlation or mediation analyses, can help to address these constraints. Genetic variation affecting the level of *Ifi203* expression was predicted to correlate positively with both hepatic PC and plasma insulin levels. This relationship was validated experimentally for hepatic PC, where reduction in liver *Ifi203* expression via adenovirus led to increased total PC levels. The fact that we only observed a trend toward increased plasma insulin concentration after reduction in liver *Ifi203* expression might be explained by limited time of gene knockdown by adenoviral treatment or degree of *Ifi203* knockdown. An accompanied increase in *Pemt* gene expression suggested that *Ifi203* plays a role in regulation of other genes important for conversion of PE to PC. Although little is known about the conserved function of *Ifi203,* it has been described to be highly expressed in liver and its expression was

shown to be suppressed during liver regeneration (Zhang *et al*, 2008). The *Ifi203* gene belongs to a large family of transcriptional suppressors, characterized by their responsiveness to interferon gamma (Landolfo *et al*, 1998). Overexpression of interferon gamma via AAV has been shown to suppress markers of hepatic fibrosis (Chen *et al*, 2005), where changes in hepatic lipidome could offer a mechanistic link. Given that many Ifi genes are also locally regulated by SNPs in this locus and that other candidates were not available in expression arrays, we cannot exclude that other candidates than Ifi203 also affect hepatic PC levels.

We also identified a locus on chromosome 11 predicted to affect the levels of several different TAG species. These data were paired with associations of hepatic gene expression and prioritized *Map2k6* as a strong candidate gene. We experimentally validated the impact of *Map2k6* on hepatic TAG levels, where hepatic overexpression of *Map2k6* lowered total TAG levels. Two other candidate genes potentially regulated by this locus, *Abca5* and *Abca6,* might be transporters of lipids (Albrecht & Viturro, 2007). Unfortunately, *Abca5* and *Abca6* were not present on the liver expression arrays. *Map2k6* phosphorylates and activates p38 MAP kinase in response to different stimuli, such as inflammation (Sabio & Davis, 2014). In accordance with these observations, others have shown that murine livers with increased levels of TAG also show lower *Map2k6* expression (Chung *et al*, 2015). One recent study showed that mice lacking *Map2k6* were protected against HF-induced obesity, possibly due to increased energy expenditure and higher *Ucp1* expression in adipose tissue (Matesanz *et al*, 2017). In contrast, we found that hepatic overexpression of *Map2k6* reduced adiposity, and plasma glucose and insulin, indicating that liver regulation of *Map2k6* may be pivotal for metabolic disease development. It is likely that the effects that we observed of *Map2k6* on hepatic TAGs and plasma insulin and glucose are at least partly explained by reduced adiposity. Future studies focused on the role of *Map2k6* in different metabolic tissues are needed to understand how this canonical pathway affects metabolic homeostasis. Given that MAPK signaling has been implicated in regulating nearly every cellular process, further efforts deconvoluting how a single canonical pathway interconnect complex metabolic processes will be crucial to integrating impacts into whole-body physiology.

The gut microbiome has a dynamic role in the regulation of inflammation and liver steatosis (Kolodziejczyk *et al*, 2019; Yuan *et al*, 2019). Changes in the gut microbial community can enhance

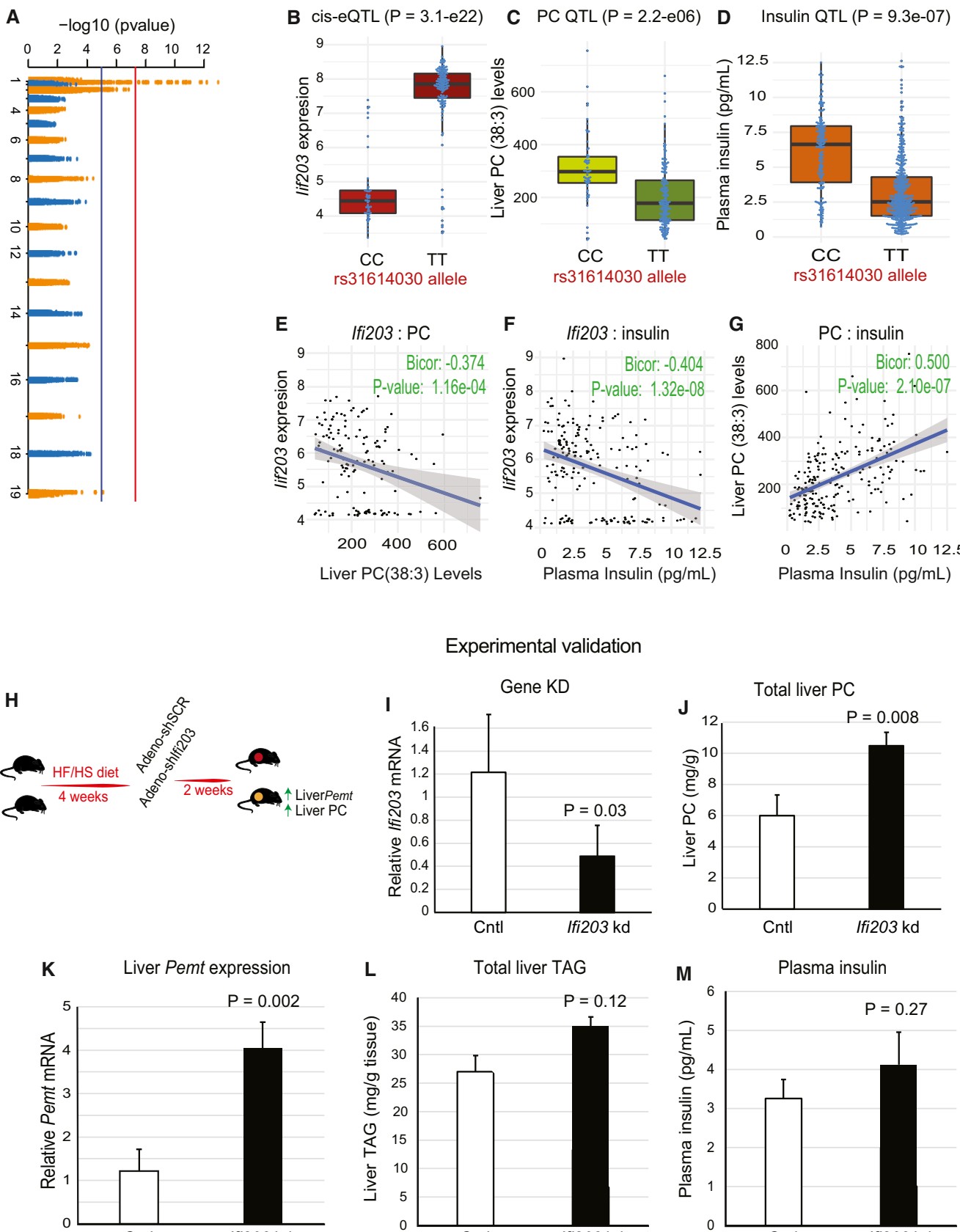

**Figure 6.**

◀

**Figure 6. Ifi203 regulates hepatic PC levels.**

A   Manhattan plot of genome-wide association for expression of *Ifi203* in liver. Red line shows Bonferroni-corrected threshold, and blue shows an FDR = 0.01 *P*-value of significance calculated based on FaST-LMM *P*-values.

B–D   Allelic variation plots showing the peak SNP for *Ifi203* expression (rs31614030) at the CC or TT allele (*x*-axis) plotted against expression of Ifi203 (B), levels of hepatic PC(38:3) (C), and plasma insulin levels (D). Red line shows Bonferroni-corrected threshold, and blue shows an FDR = 0.01 *P*-value of significance calculated based on FaST-LMM *P*-values.

E–G   Correlation between the parameters listed above showing significant relationships between hepatic Ifi203 and PC(38:3) (E), hepatic *Ifi203* and plasma insulin levels (F) or PC (38:3) and plasma insulin (G). *P*-values were calculated based on significance of regression (Student's test) and adjusted for multiple comparisons (FDR = 0.05).

H   Experimental design for validation of Ifi203 as a regulator of total hepatic PC levels on a HF/HS diet.

I–M   Mice receiving the control virus (open bars) or shIfi203 (black bars) were analyzed for liver expression of *Ifi203* (I), total PC levels in liver (J), expression of *Pemt* (K), total liver TAG content (L), or plasma insulin levels (M). *P*-values calculated using a Student *t*-test between groups. Data represent means ± SEM (*n* = 4–5 per group).

the severity of NAFLD via microbiome-derived metabolites (Kolodziejczyk *et al*, 2019). Gut microbes can utilize carbohydrates to synthesize different short chain fatty acids that can regulate host metabolism. Short chain fatty acids can directly act as lipid precursors in the liver or mediate other effects by acting as ligands of G protein-coupled receptors (Marra & Svegliati-Baroni, 2018). For example, a recent study showed that an unfavorable gut microbiome metabolite production is sufficient to induce hepatic steatosis in normal mice (Yuan *et al*, 2019). Furthermore, another recent study showed that a gut microbiome alcohol production is sufficient to induce hepatic steatosis in normal mice (Yuan *et al*, 2019). The composition of the gut microbiome is highly heritable (Org *et al*, 2015), suggesting that host genetic composition exerts a striking control over the function of the microbiota. Systems genetics approaches such as those shown here offer tools to examine such relationships. We provide several notable correlations between types of microbes and hepatic lipids. For example, *Anaeroplasma, AF12,* and *Desulfovibrio* show negative correlations with many CL and LPC species. CLs are essential for mitochondrial bioenergetics functions (Maguire *et al*, 2017) and are involved in the development of hepatic steatosis (Jha *et al*, 2018a). LPC is a phospholipid generated from PC by the removal of one of the fatty acid groups and might mediate lipotoxicity in hepatocytes (Hirsova *et al*, 2016). Because *Desulfovibrio* increases in the gut when C57BL/6J mice transition into hepatic steatosis and NASH (Xie *et al*, 2016), it might be speculated that *Desulfovibrio* plays a role in NAFLD progression by affecting hepatic CL and LPC levels. These new relationships require direct experimentation to prove directionality and causality.

Our study has several limitations. Our association analyses between hepatic lipids and phenotypes or gut bacterial species are hypothesis generating. Follow-up studies are required to support causal relationships. Because the members of each lipid category are largely correlated, it is important to interpret single correlations between lipid species and traits with caution. Furthermore, the number of mice assayed in each strain ranged from 1 to 4 individuals. In particular, only one mouse was used for two strains, which could contribute to analysis bias. Although we provided experimental *in vivo* evidence in mice for our murine GWAS candidate genes *Ifi203* and *Map2k6* as regulators of accumulation of specific classes of liver lipids, we did not both overexpress and knock down the genes in mice. We also did not test whether we could get a dose-dependent effect on hepatic lipids with different concentrations of adenoviruses and AAVs. For example, we cannot exclude that less overexpression of Map2k6 would resulted in a different phenotype.

In summary, our results provide data for hypothesis generation for genetic and environmental factors of key importance for the regulation of hepatic lipids in diet-induced NAFLD. We provide several different examples of how the utilization of systems genetics approaches can be applied to discover links between the hepatic lipidome and phenotypic traits, and identified and validated two novel regulators of hepatic lipids.

# Material and Methods

### Animals

All animal experiments were approved by the University of California Los Angeles (UCLA) Animal Care and Use Committee, in accordance with Public Health Service guidelines. Mice strains in the HMDP study were obtained from the Jackson laboratory and have been described in detail (Hui *et al*, 2015). Experimental design of the high-fat/high-sucrose (HF/HS) feeding study has also been described previously (Parks *et al*, 2013; Hui *et al*, 2015). Briefly, the mice were maintained on a chow diet (Ralston Purina Company) until 8 weeks of age before switching to a HF/HS (Research Diet-D12266B, New Brunswick, NJ) diet for another 8 weeks. Mice were housed in a 12-h light/dark cycle with *ad libitum* feeding. Mice in the diet study were either maintained on chow diet for 16 weeks or maintained on chow diet until 8 weeks of age, and switched to a HF/HS diet for 8 weeks. Mice from both studies were euthanized after 4-h fasting starting between 10:30 AM and noon.

### Lipid extraction and quantification

Liver samples (about 20 mg) from 279 male mice (*n* = 1–4 mice per strain; *n* = 1 in 2 strains; *n* = 2 in 23 strains; *n* = 3 in 73 strains; *n* = 4 in 3 strains) were homogenized and lipids extracted by 10 vol of chloroform:methanol (*v:v* 2:1) (Folch *et al*, 1957). Internal standards, one for each lipid group, were added to the murine liver samples (1 µg/ml) prior to adding the extraction solvent. The following lipids were used as internal standards: PC-28:0, PE-28:0, PG-30:0, PA-28:0, PI-31:1, Cer-35:1, SPM-35:1, DAG-28:0, TAG-39:0, CE-15:0, CL-56:0, LPC-17:1, LPE-15:0, and FFA-17:1.

Lipidomics analyses were performed on mouse liver extracts using high-performance liquid chromatography (HPLC) coupled to time-of-flight mass spectrometry (TOF-MS) as previously described (Norheim *et al*, 2018). This platform allows determination of

glycerolipids, glycerophospholipids, CL, sphingolipids, FFA, and CE. In total, 256 specific lipids within these classes were identified. A 1260 Agilent chromatographic system comprising an auto-sampler, a binary pump, and a TCC column heater unit coupled to a time-of-flight mass spectrometer with Agilent JetStream ionization module for enhanced sensitivity was operated in both positive and negative ionization modes. To obtain high-resolution chromatographic separation of the lipids, a C18-XB Kinetex analytical column with (2.1 × 150 mm, 2.6 μm) was used with a flow rate of 0.8 ml/min. The eluting mobile phase was generated using A (10:90 $v/v$, acetonitrile: 10 mmol/l ammonium formate) and B (70:25:5 $v/v$, isopropanol: acetonitrile, 10 mmol/l ammonium formate) mixed by a binary pump generating a mobile phase gradient as follows: 0 min (50% B), 12 min (70% B), 55 min (100% B), and 65% (100% B). Injected volume was 5 μl (positive mode) and 10 μl (negative mode).

Measurements of total lipid content using colorimetric assays were performed as previously described (Norheim *et al*, 2017). A colorimetric assay from Sigma (St. Louis, MO, USA) and Wako (Richmond, VA, USA) was used to quantify TAG and PC, respectively. Total cholesterol and unesterified cholesterol were measured as described previously with in-house reagents (Castellani *et al*, 2008). All raw lipidomics data are provided in Dataset EV8. Integrated results from lipidomics and other datasets (e.g., mapping, correlation structure) are available at https://systems.genetics.ucla.edu/.

## Adenoviral construction and administration

Recombinant adenovirus was generated using the AdEasy system (Bennett *et al*, 2013). Briefly, linearized shuttle vector containing full-length mouse cDNA for *Ifi203* was transformed into *Escherichia coli* BJ5183AD cells containing the adenoviral backbone plasmid pAdEasy-1 for homologous recombination. Positive recombinants were linearized and transfected into HEK293AD cells for virus packaging and propagation. Adenoviruses expressing the candidate gene were purified by CsCl banding and stored at −80°C until use. For adenoviral infection, 10-week-old male C57BL/6J mice (fed a HF/HS diet for 4 weeks) were injected the adenoviral construct (∼2.5 × 10⁹ PFUs diluted in 0.2 mL saline) intraperitoneally. After overnight fasting, mice were sacrificed 9 days after injection, tissues were extracted, and gene expression was assessed by RT–PCR. The control group included mice injected with adenoviral construct expressing the GFP gene.

## AAV vector construction and *in vivo* transduction

The mouse and mitogen-activated protein kinas kinase 6 (Map2k6) open reading frame was PCR-amplified from Dharmacon cDNA clone ID 30541969 and cloned into the pENN.AAV.TBG.PI.eGFP vector (p1014; Penn Vector Core) to replace eGFP. This vector drives transgene expression from the liver-specific thyroxine-binding globulin (TBG) promoter (Yan *et al*, 2012). AAV serotype 8 (AAV8) particles were packaged and purified on a fee-for-service basis at the Penn Vector Core (Perelman School of Medicine, University of Pennsylvania, USA). eGFP-expressing vector was used as control. AAV8 particles were intraperitoneally injected at a dose of 3 × 10¹² gc per mouse in 8-week-old male mice. After injection,

the mice were switched to a HF/HS diet for 8 weeks. Western blotting was used to verify overexpression of hepatic Map2k6.

## RNA extraction and reverse transcription

Cells or tissue were homogenized in Qiazol (Qiagen), and RNA extraction was carried out as recommended. Samples were suspended in 0.5 ml Qiazol each; then, 100 μl chloroform was added. After vortexing, phase separation was achieved with centrifugation at 18,000 $g$ for 15 min. The aqueous layer was then transferred to 1 ml isopropanol, vortexed, and then centrifuged again. The remaining pellets were washed in 70% ethanol in water then air-dried following centrifugation for 10 min. Purified RNA was then suspended in 30 μl of water and assessed for purity and concentration using a NanoDrop ND-100 Spectrophotometer. 2 μg of total RNA per sample was reverse transcribed using a High-Capacity cDNA reverse transcription kit (Applied Biosystems) with random primers. Reverse-transcribed cDNA was then diluted in water for qPCR analysis.

## Quantitative PCR

Quantitative PCR was carried out using a Kappa SYBR Fast qPCR Kit as recommended by the manufacturer. Samples were analyzed on a LightCycler 480 II (Roche) and using the Roche LightCycler 1.5.0 Software. All qPCR targets were normalized to geometric mean of RPL13a and PPIA expression and quantified using the delta Ct method. All qPCR primer sequences were obtained from Primer-Bank (http://pga.mgh.harvard.edu/primerbank).

## Microbial DNA extraction and sequencing

Cecum samples were collected (Parks *et al*, 2013) and sequenced (Org *et al*, 2015) in previous studies, and methods are briefly described here. Microbial DNA was extracted following MO BIO PowerSoil®-htp 96 Well Soil DNA Isolation Kit. The 16S rRNA V4 hypervariable region was amplified with barcoded primers (Caporaso *et al*, 2011) in triplicate using the 5 PRIME HotMasterMix (VWR). Products were quantified with Quant-iT™ PicoGreen® dsDNA Assay Kit (Thermo Fisher), and samples were combined in equal amounts (∼ 250 ng per sample) to be purified with the Ultra-Clean PCR® Clean-Up Kit (MO BIO). Pooled amplicons were sequenced on the Illumina MiSeq platform.

Raw sequences were processed with the open source Quantitative Insights Into Microbial Ecology (QIIME) software package version 3.6.1 (Caporaso *et al*, 2010). Sequences were binned at 97% similarity using UCLUST against a Greengenes reference database (McDonald *et al*, 2012). Singletons, OTUs representing less than 0.005% total relative abundance, and unsuccessful samples with less than 1,000 reads were removed resulting in 14,722,208 total reads, with an average of 23,258 reads per sample. Sequences were rarefied to 10,056 reads per sample to accommodate unequal sampling depth leaving 613 samples for downstream analyses.

## Plasma insulin, glucose, and lipids

Blood was collected from mice using retro-orbital bleeding under isoflurane anesthesia. Plasma levels of insulin, glucose, HDL, and

LDL were measured as reported previously (Castellani *et al*, 2008). Homeostatic model assessment of IR (HOMA-IR) was calculated using the equation [(glucose × insulin)/405].

### Western blotting

Western blotting was performed as described previously (Chella Krishnan *et al*, 2018; Seldin *et al*, 2018). Primary antibodies were used as follows mouse monoclonal Map2k6 (Abcam # ab33866) and rabbit monoclonal β-actin (Cell Signaling # 4967S, 1:1,000). Blots were imaged using IMAGER.

### Association analysis

Genotypes for the mice strains were obtained from the Jackson Laboratories using the Mouse Diversity Array (Yang *et al*, 2009). Single nucleotide polymorphisms (SNP), which had poor quality or had a minor allele frequency (MAF) of less than 5% and a missing genotype rate of less than 10%, were removed. After filtering, 200,000 SNPs were left. Genome-wide association for hepatic lipids was performed using Factored Spectrally Transformed Linear Mixed Models, which applies a linear mixed model to correct for population structure (Lippert *et al*, 2011). A cutoff value for genome-wide significance was set at $3.46 \times 10^{-6}$, as determined previously for the HMDP (Bennett *et al*, 2010). Hepatic lipids not identified in more than 50% of the strains were excluded from the analysis (36 lipid species). LD was determined by calculated pairwise $r^2$ SNP correlations for each chromosome. Approximate LD boundaries were determined by visualizing $r^2 > 0.8$ correlations in MATLAB (MathWorks).

### Accession numbers

The NCBI GEO accession number for microarray data reported in this paper is GSE64770. Microbiota composition data are available via NCBI Sequence Read Archive (SRA; http://www.ncbi.nlm.nih. gov/sra/) under accession number SRP059760.

### Lipidome module construction

Hepatic lipidome measures were collapsed into modules using WGCNA. Briefly, hierarchical clustering was used to detect outliers, retaining 211 lipids. Next, blockwise module construction was performed using a minimum module size of five lipids and a merge cut height of 0.25.

### Statistics

Correlations were calculated with biweight midcorrelations from the R package WGCNA (Langfelder & Horvath, 2008). Unless otherwise noted, values are expressed as means ± SEM. The two-sample Student's t-test was used to evaluate the difference between the two groups. Identification of differentially expressed lipid species in each condition was performed using the R package limma (Ritchie *et al*, 2015). All analyses were performed using R 3.5.3 (Vienna, Austria), and *P*-values < 0.05 were considered statistically significant. Network models were visualized using the package qgraph and manhattan plots generated using qqman.

## Data availability

The NCBI GEO accession number for microarray data reported in this paper is GSE64770 (http://www.ncbi.nlm.nih.gov/geo/query/acc.cgi?acc=GSE64770). Microbiota composition data are available via NCBI Sequence Read Archive (SRA; http://www.ncbi.nlm.nih. gov/sra/) under accession number SRP059760. Lipidomics data are provided in Dataset EV8.

Expanded View for this article is available online.

### Acknowledgements

We thank Hannah Qi and Zhiqiang Zhou for expert assistance with mouse experiments, and Sarada Charugundla for plasma metabolite analysis. This work was supported by the Norwegian Research Council 240405/F20 (F.N), NIH grants: DK097771 (M.M.S), T32HL69766 (M.M.S), HL138193 (M.M.S), HL28481 (A.J.L), HL30568 (A.J.L), HL144651 (A.J.L), DK117850 (A.J.L), NIH-K99 DK120875 (K.C.K), and American Heart Association postdoctoral fellowship 18POST33990256 (K.C.K).

### Author contributions

FN, KCK, TB, LV, JL, MM, TEG, MP, KR, CAD, STH, AJL, and MMS, designed experiments. FN, KCK, TB, LV, CP, BWP, YM, JL, JAW, MM, MP, STH, and AW performed the experiments. FN, KCK, CP, KTD, AJL, and MMS analyzed raw data. FN, CAD, AJL, and MMS wrote the manuscript, which was reviewed by all authors.

### Conflict of interest

T.B, T.E.G, and C.A.D are affiliated with Vitas Ltd. T.B and T.E.G are employed and stock owners, whereas C.A.D is board member, stock owner, and consultant in Vitas. Vitas Ltd. performed the lipidomics analyses. The other authors declare no conflict of interests.

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
