## [Review Process File · Molecular Systems Biology]

Genetic regulation of liver lipids in a mouse model of insulin resistance and hepatic steatosis

Frode Norheim, Karthickeyan Chella Krishnan, Thomas Bjellaas, Laurent Vergnes, Calvin Pan, Brian Parks, Yonghong Meng, Jennifer Lang, James Ward, Karen Reue, Margarete Mehrabian, Thomas Gundersen, Miklos Peterfy, Knut Dalen, Christian Drevon, Simon Hui, Aldons Lusic, and Marcus Seldin

DOI: 10.15252/msb.20209684

Corresponding author(s): Marcus Seldin (mseldin@uci.edu) , Aldons Lusic (jlusic@mednet.ucla.edu)

Review Timeline:	Submission Date:	4th May 20
	Editorial Decision:	4th Jun 20
	Revision Received:	1st Sep 20
	Editorial Decision:	20th Oct 20
	Revision Received:	3rd Nov 20
	Accepted:	10th Nov 20

Editor: Jingyi Hou

Transaction Report:

Thank you for submitting your work to Molecular Systems Biology. We have now heard back from the three reviewers who agreed to evaluate your manuscript. As you will see below, the reviewers acknowledge the potential interest of the presented findings. They raise however a series of concerns, which we would ask you to address in a major revision.

I think that the reviewers' recommendations are rather clear and there is therefore no need to reiterate the comments listed below. In light of the concerns of reviewer #3, we would ask you to edit the manuscript to make sure that the main findings are sufficiently clear and easily accessible to the general audience of Molecular Systems Biology.

All other issues raised by the reviewers need to be satisfactorily addressed as well. As you may already know, our editorial policy allows in principle a single round of major revision and it is therefore essential to provide responses to the reviewers' comments that are as complete as possible.

On a more editorial level, please do the following.

REFEREE REPORTS

Reviewer #1:

Norheim et al. investigate the genetic architecture of liver lipids in a mouse model of hepatic

steatosis. Overall, this is interesting and well-presented work, and the data generated will be a valuable resource for the lipidomics research community.

Specific comments

1. Authors used 559 mice (102) strains. However, description of how many mice were included in different experiments is missing. For example, authors selected three strains that responded differently to HF/HS diet, then measured hepatic lipidomes in these strains (fed HF/HS or normal chow diet). How many mice were in each group?
2. In Figure 1B, authors show heatmap of the fold-change (log₂) of each lipid in HF/HS as compared to chow diet. Figure 1C suggests that not all of these lipids were changing. Therefore, it would be useful to mark in Figure 1B (like in Figure 7B), which lipids were statistically significant, and provide individual lipid name alongside, as authors have already highlighted few of them.
3. Author suggest that genetic diversity causing variation in several sphingolipids (Cer-PE and Cer-PC) species were linked to traits such as plasma insulin, however, while looking at the Figure 3B, this is not obvious. Could author clarify it and point to this finding in the figure or table?
4. Author mention that the relative genetic variation in hepatic lipidome is higher among the less abundant lipids. Is there any explanation for this finding? Could this be related to analytical variation?
5. Association between microbiome and lipids is an interesting aspect. Could author identify and discuss any particular lipids that may be microbiome related, directly or indirectly?

Reviewer #2:

In this study, the authors investigated the association between genetic variations and lipidomics, phenotypic traits in NAFLD using around 100 mice strains. They also validated the causal effect of two key genes they identified in additional mouse experiment. I find the validation sections particularly impressive. However, I think the manuscript still need to be improved before acceptance. My detailed comments are as below:

1. 'These results suggest that genetic differences have equal, if not greater impacts on the hepatic lipidome as compared to that of a HF/HS diet alone, at least in the context of these three HMDP strains' I believe there are much better ways (e.g. variance analysis) to compare the effect of diet and genetic differences on the lipidomic changes than Figure 1C.
2. In the manuscript the authors wrote 'As examples, ceramides correlated with plasma glucose levels and phosphatidylethanolamine (PE) correlated with body fat percentage (Fig. 3B)', but Figure 3B does not clearly support it.
3. 'When combined, these lipids showed significant correlations with body weight as well as plasma insulin and hepatic triglyceride levels (Fig. 3C)', there is no hepatic triglyceride levels in figure 3c or the author did not clarify its legend.
4. It is very unclear why the author selected Map2k6 and Irf203 among the 76 genes they identified to validate. Please justify the selection.
5. 'This gene has been associated with unfavorable lipid profiles in humans' I believe it is 'species' rather than 'gene', or the authors need to clarify it.
6. The 'Relationships between gut microbiota and hepatic lipids' section needs to be extended to at least give a biological message that is novel as the current version only focuses on the validation of

previous conclusions.

7. There is no information related to microbiota analysis of the study at the material methods section.

8. Lipid species clustered into 12 modules with colour code. How is this discrimination performed? A table should also be provided that shows the lipid species in each module. (not only numbers of lipids within a class per module)

9. On page 11... For adenoviral infection, mice fed a HF/HS diet for 4 weeks were injected the adenoviral construct. After 9 days of injection mice were sacrificed. What is the basement of this experimental design?

10. In Figure 7 b, there are two data of the Ruminococcus genus, and these data are different from each other. It is recommended to rearrange this heatmap and organize the taxonomic levels in the x axis.

11. On page 7... Authors directed the readers to the Supplemental Figure 4 for the relationship between microbiome abundance profiles and lipid modules. Findings should be written briefly in the results section.

12. On page 4... The most abundant lipid class (TAG) accounted for 44 to 79% of total lipids in the liver and the content of phosphatidylcholine (PC) varied > 3-fold (Fig. 2B). Outliers might be highlighted here. What are the mean values?

13. On page 9... Discussion about the gut microbiome and lipids has only general info and a few example articles from the literature. This paragraph discusses absolutely nothing with the study findings.

14. In the results section at paragraph 1, authors wrote that 252 lipid species were detected but the sum of the species at the venn-diagram of figure1-c is 251. I assume there is a typo.

Reviewer #3:

The manuscript from Norheim et al. describes a systems genetics approach aimed to identify novel modulators of the hepatic lipidome in the context of NAFLD.

While both the approach and the topic are very timely, the manuscript is only descriptive and fails to deliver a clear message to the reader. They might be able to improve the flow of the story by re-organizing the content. This reviewer would suggest starting with high-level analysis, then data integration like QTL analysis and eventually experimental validation. Some panels (like Fig 2C, D) may have to be moved to the supplementary materials. There are also a few panels/analyses that do not help the storyline directly.

Major comments:

1. The authors described that they used 268 male mice in the Methods section, but listed 559 mice in Fig. 2A. How many mice were used exactly?

2. How did the authors compute the lipid changes with genetics in Fig. 1C? The method part describing the statistics is not very clearly written.

3. In general, the method part does not seem complete. The authors need to explain their methodology with more details.

4. There are some computational methods to quantify the proportion of variance explained by a given factor like diet. They can use those methods instead of what they did for Fig 1C.

5. The schemes of the figure (for example Fig. 2A, 3C) are very confusing.

6. The authors focus their attention on the positive correlation between hepatic TAG levels and body weight or plasma insulin (Fig. 3C) but this is not the strongest correlation result. Why this choice?

7. The authors also stated that the members of each lipid category are largely correlated, if so, how did they consider potential confounding factors for the follow-up analysis using correlation?
8. Since the authors have generated a new lipidomics dataset in the context of this manuscript, why did they use the previous GWAS significance threshold from Bennett et al, 2010?
9. If the association between pex16 and several LPCs was demonstrated through a genetic marker, why did they perform correlation analysis in Fig 4A? Alternatively, they could apply a causal inference method.
10. The authors picked Map2k6 as the candidate regulator for TAGs in Fig. 5, how was this gene prioritized over the nearby Abca genes (potential transporters of lipids, ref: PMID: 16586097)?
11. What is the GWAS result for PC(C38:3) in Fig. 6? Can the authors provide the Manhattan and regional plots? Also, it is not as the authors claimed that lfi203 is the most proximal gene of the peak SNP (the peak SNP rs31614030 is located at chr1 - 173,308,737, while the lfi203 gene is located at chr1 - 173,920,400). There are quite a few genes between the peak SNP and lfi203 gene, for example, Aim2 and a cluster of other lfi genes. Why did the authors pick lfi203 over the others? Is there any connection/co-regulation between lfi203 and other lfi genes?
12. The validation experiments are weak. First of all, the timeline of the experiments, as well as the AAV approach used to assess the biological relevance of Map2k6 (Fig. 5) and lfi203 (Fig. 6) are different. The hepatic overexpression of Map2k6 does not affect plasma lipid levels: how do the authors explain the effect seen on body fat and plasma glucose/insulin levels? Along the same lines, the adenovirus system used to reduce lfi203 expression is not liver-specific and the metabolic effects observed are not significant (Fig. 6L and 6M).
13. Fig. 5F why is there no endogenous Map2k6 detected in the AAV-GFP mice? Is it because the levels seen in the overexpressed group are so high that the endogenous one becomes proportionally undetectable? If so, this really questions the validity of an experiment where the gene of interest is overexpressed at supra-physiological levels...The authors might want to lower the concentration of virus used.
14. The analysis of the gut microbiome and its link with the liver lipidome is very superficial and does not add much to this manuscript. The same is true for the transcriptome analysis of the other metabolic organs (Fig. 7C and 7D).
15. In Fig. 7D, the authors used a method they previously described to study the endocrine circuits using gene expression data across tissues. However, the values in Fig. 7D are very different from the ones in their original paper, where the final scores were at most 2~5 (e.g. Fig. 2 in Seldin 2018). Why are these scores so different? This method uses gene expression and not secretomics data. How valid are these results?

Minor comments

1. Text in many figures is squeezed.
2. There are many typos in the text and figures.
3. Module color labels in Fig3C and FigS2 are different.
4. The sample size is not mentioned for many figures, for example, Fig5G-K.
5. Fig 4D is missing.

Reviewer #1:

Norheim et al. investigate the genetic architecture of liver lipids in a mouse model of hepatic steatosis. Overall, this is interesting and well-presented work, and the data generated will be a valuable resource for the lipidomics research community.

Answer: Thank you for the positive feedback and suggestions which improved our manuscript.

Specific comments

1. Authors used 559 mice (102) strains. However, description of how many mice were included in different experiments is missing. For example, authors selected three strains that responded differently to HF/HS diet, then measured hepatic lipidomes in these strains (fed HF/HS or normal chow diet). How many mice were in each group?

Answer: We apologize for the confusion. This was due to the fact that 559 mice have been assayed for different molecular and clinical traits. Lipidomics was measured in 279 mice (102 strains, n = 1-4 mice per strain) and we have corrected texts/legends accordingly. For the measures of dietary impact (Fig 1C), we used n = 3 mice per strain and diet group when we compared the HF/HS vs normal chow. This information has also been added to the text and the legends.

2. In Figure 1B, authors show heatmap of the fold-change (log2) of each lipid in HF/HS as compared to chow diet. Figure 1C suggests that not all of these lipids were changing. Therefore, it would be useful to mark in Figure 1B (like in Figure 7B), which lipids were statistically significant, and provide individual lipid name alongside, as authors have already highlighted few of them.

Answer: Many lipids achieved significance (adjusted pvalue <0.001) for the comparison, so upon adding, the figure became quite messy. That being said, we agree that knowing the species would be helpful. Therefore, we have added all of the differential expression results based on diet to a new supplemental table (**Dataset EV 1**). We have also highlighted some liver lipids from the heatmap in a separate figure (**Figure 1C**), to provide examples of liver lipids within the same class that are differently regulated by diet. These lipids have been marked for statistical significance.

3. Author suggest that genetic diversity causing variation in several sphingolipids (Cer-PE and Cer-PC) species were linked to traits such as plasma insulin, however, while looking at the Figure 3B, this is not obvious. Could author clarify it and point to this finding in the figure or table?

Answer: Thank you for pointing this out. We have changed the sentence in the Discussion to: "For example, genetic diversity causing variation in several Cer-PE species were linked to traits such as body weight and HOMA-IR." Page 8, line 309

4. Author mention that the relative genetic variation in hepatic lipidome is higher among the less abundant lipids. Is there any explanation for this finding? Could this be related to analytical variation?

Answer: The reason that the relative genetic variation is higher among the less abundant lipids is that you can measure lipids that are higher level more accurately than lipids that are lower level. So the reviewer is correct in that analytical variation plays a role. We have added the following to the results: "While analytical variation can clearly contribute to these observations, higher variation among lower abundances across genetic backgrounds has been widely appreciated for multiple omics measures and reviewed in detail (ref: PMID: 27104977)." Page 4, line 145

5. Association between microbiome and lipids is an interesting aspect. Could author identify and discuss any particular lipids that may be microbiome related, directly or indirectly?

Answer: We have now added more information about the associations between the microbiome and lipids in both the result and discussion section. Specifically, we have expanded discussion of correlations to individual lipids and to lipid modules as a whole. We highlight two new relationships between liver lipids and the gut microbiome. Page 5, line 150

Reviewer #2:

In this study, the authors investigated the association between genetic variations and lipidomics, phenotypic traits in NAFLD using around 100 mice strains. They also validated the causal effect of two key genes they identified in additional mouse experiment. I find the validation sections particularly impressive. However, I think the manuscript still need to be improved before acceptance. My detailed comments are as below:

Answer: Thank you for the positive feedback, and your help in improving our manuscript.

1. 'These results suggest that genetic differences have equal, if not greater impacts on the hepatic lipidome as compared to that of a HF/HS diet alone, at least in the context of these three HMDP strains' I believe there are much better ways (e.g. variance analysis) to compare the effect of diet and genetic differences on the lipidomic changes than Figure 1C.

Answer: This point was mentioned by two reviewers. Given the smaller size and specific context of these comparisons, we felt that the analysis presented more confusion than helped to show the impact of genetics and diet. We therefore, removed panel C from figure 1 and toned-down our discussion of genetic vs dietary contributions to hepatic lipids. In the revised manuscript, we provide the impacts of diet on lipidome as a strong basis that our studies recapitulate previously-described pathways, but shifted focus to other analyses.

2. In the manuscript the authors wrote 'As examples, ceramides correlated with plasma glucose levels and phosphatidylethanolamine (PE) correlated with body fat percentage (Fig. 3B)', but Figure 3B does not clearly support it.

Answer: Thank you for pointing this out. The sentence has been changed to: “As examples, the levels of some hepatic ceramides and phosphatidylethanolamines (PE) correlated negatively with plasma glucose levels and body fat percentage, respectively (Fig. 3B)” Page 6, line 198

3. 'When combined, these lipids showed significant correlations with body weight as well as plasma insulin and hepatic triglyceride levels (Fig. 3C)', there is no hepatic triglyceride levels in figure 3c or the author did not clarify its legend.

Answer: We have changed to sentence into: “When combined, this module of lipids showed significant correlations with body weight as well as plasma insulin and HDL (Fig. 3C)” Page 6, line 215

4. It is very unclear why the author selected Map2k6 and Ifi203 among the 76 genes they identified to validate. Please justify the selection.

Answer: We agree that we could have chosen to follow up other genes as well. We selected Map2k6 because it seemed to be a good candidate to affect hepatic steatosis (hepatic TAGs) and mentioned in the results: “Given the clear role of TAG accumulation in hepatic steatosis, we searched for genomic regions which associated with multiple TAG species. We observed that TAG(56:3), TAG(54:4),

TAG(48:2), TAG(48:1) and TAG(48:0) all mapped to approximately the same area on chromosome 11 (**Fig. 5A, Dataset EV2**). This locus included only three potential candidate genes: ATP binding cassette subfamily A member 5 (*Abca5*), ATP binding cassette subfamily A member 6 (*Abca6*), and mitogen-activated protein kinase 6 (*Map2k6*). Integration with hepatic gene expression revealed that *Map2k6* was regulated in *cis* by the same loci (**Fig 5B**) Page 7, line 245

Our rationale for selecting *Ifi203* was both that 1) The association between the peak snp for multiple PC species (rs31614030) and *Ifi203* was stronger than those compared to any other gene within a 1Mb window and 2) This locus was among the only regions which achieved GWA bonferroni significance for the gene, multiple lipids and clinical trait (Insulin). We have added the following to the results section: “ We note that this was the only example where a locus co-mapped to expression of a gene in *cis*, multiple lipid species and a clinical trait. Significant correlations were also observed between all of these measures in directions consistent with the effect of the SNP. While other genes (including interferon-activatable family members) within the same locus showed strong associations to the peak SNP, albeit not as significant, *Ifi203* was the only one which also correlated in directions consistent with the genetic effects “ Page 7, line 273

5. 'This gene has been associated with unfavorable lipid profiles in humans' I believe it is 'species' rather than 'gene', or the authors need to clarify it.

Answer: We have now changed the sentence into: “*Anaeroplasma* has been associated with unfavorable lipid profiles in humans.” Thank you for catching this error. Page 5, line 164

6. The 'Relationships between gut microbiota and hepatic lipids' section needs to be extended to at least give a biological message that is novel as the current version only focuses on the validation of previous conclusions.

Answer: We have updated the figures, results and discussion sections to provide a biological messages. We have moved the microbiota analyses to the top (**Fig 2**) and expanded our discussion. Specifically, we have expanded discussion of correlations to individual lipids and to lipid modules as a whole. We highlight two new relationships for mechanisms of gut-host communication our data suggests.

7. There is no information related to microbiota analysis of the study at the material methods section.

Answer: We apologize for this oversight and have now added the information: “Cecum samples were collected (Parks *et al.*, 2013) and sequenced (Org *et al.*, 2015) in previous studies, and methods are briefly described here. Microbial DNA was extracted following MO BIO PowerSoil[®]-htp 96 Well Soil DNA Isolation Kit. The 16S rRNA V4 hypervariable region was amplified with barcoded primers (Caporaso *et al.*, 2011) in triplicate using the 5 PRIME HotMasterMix (VWR). Products were quantified with Quant-iT[™] PicoGreen[®] dsDNA Assay Kit (Thermo Fisher) and samples were combined in equal amounts (~250 ng per sample) to be purified with the UltraClean PCR[®] Clean-Up Kit (MO BIO). Pooled amplicons were sequenced on the Illumina MiSeq platform.

Raw sequences were processed with the open source Quantitative Insights Into Microbial Ecology (QIIME) software package version 3.6.1 (Caporaso *et al.*, 2010). Sequences were binned at 97% similarity using UCLUST against a Greengenes reference database (McDonald *et al.*, 2012) Singletons, OTUs representing less than 0.005% total relative abundance, and unsuccessful samples with less than 1,000 reads were removed resulting in 14,722,208 total reads, with an average of 23,258 reads per sample. Sequences were rarefied to 10,056 reads per sample to accommodate unequal sampling depth leaving 613 samples for downstream analyses” Page 12, line 501

8. Lipid species clustered into 12 modules with colour code. How is this discrimination performed? A table should also be provided that shows the lipid species in each module. (not only numbers of lipids within a class per module)

Answer: We have provided more information for specific parameters for WGCNA in the manuscript (both results and methods), as well as included a new table (**dataset EV4**) showing module membership for all lipids.

9. On page 11... For adenoviral infection, mice fed a HF/HS diet for 4 weeks were injected the adenoviral construct. After 9 days of injection mice were sacrificed. What is the basement of this experimental design?

Answer: We have used this design before to study the effect of genes on hepatic steatosis. The rationale is that because the adenovirus only lasts for up to 2 weeks, we want to induce hepatic steatosis (HF/HS diet for 4 weeks) before injection of the adenoviral construct. We have now added the following statement in the result section: "Mice were fed a HF/HS diet for 4 weeks to induce hepatic steatosis, then administered an adenovirus (2×10^9 PFU/mouse) containing either a scrambled control or a shRNA targeting *Ifi203* expression under a ubiquitous CMV-U6 promoter" Page 12, line 467

10. In Figure 7 b, there are two data of the Ruminococcus genus, and these data are different from each other. It is recommended to rearrange this heatmap and organize the taxonomic levels in the x axis.

Answer: There are two taxa named Ruminococcus. There's one at the beginning and one in the middle. The one in the middle has a period before and after the name. This is because the name is actually [Ruminococcus] and I think R dropped the brackets and replaced them with periods. The bracket means that this groups of bacteria was misclassified as Ruminococcus and is awaiting its proper nomenclature.

<https://support.nlm.nih.gov/knowledgebase/article/KA-03379/en-us>. We have now changed the periods with brackets around Ruminococcus, and rearranged the heatmap

11. On page 7... Authors directed the readers to the Supplemental Figure 4 for the relationship between microbiome abundance profiles and lipid modules. Findings should be written briefly in the results section.

Answer: As mentioned above, we now highlight two new relationships for mechanisms of gut microbe-host communication our data suggests.

12. On page 4... The most abundant lipid class (TAG) accounted for 44 to 79% of total lipids in the liver and the content of phosphatidylcholine (PC) varied > 3-fold (Fig. 2B). Outliers might be highlighted here. What are the mean values?

Answer: This is a valid point. We have indicated summary statistics for each lipid class and individual lipid in the new Dataset EV2 and Dataset EV3, respectively. We have also pointed out two exceptions to this high level of variation in the results:

"Summary level statistics, such as mean abundance and variance across the 279 mice are provided for each lipid class (**Dataset EV2**) and individual lipids (**Dataset EV3**). Not all lipid species varied substantially across strains. For example, Cer(34:2) and PC(34:1), showed minimal variation relative to the mean compared to others (**Dataset EV3**)" Page 4, line 142

13. On page 9... Discussion about the gut microbiome and lipids has only general info and a few example articles from the literature. This paragraph discusses absolutely nothing with the study findings.

Answer: We have now extended the discussion section about the gut microbiome and lipids. As mentioned above, we focus on our module-microbe relationships to highlight potentially new mechanisms of microbe-host interactions.

14. In the results section at paragraph 1, authors wrote that 252 lipid species were detected but the sum of the species at the venn-diagram of figure1-c is 251. I assume there is a typo.

Answer: We apologize for the oversight. The reviewer is correct in that one lipid did not change in any parameters. We have nonetheless removed this comparison from the study, as it seemed to create more confusion than contribute to the study.

Reviewer #3:

The manuscript from Norheim et al. describes a systems genetics approach aimed to identify novel modulators of the hepatic lipidome in the context of NAFLD.

While both the approach and the topic are very timely, the manuscript is only descriptive and fails to deliver a clear message to the reader. They might be able to improve the flow of the story by re-organizing the content. This reviewer would suggest starting with high-level analysis, then data integration like QTL analysis and eventually experimental validation. Some panels (like Fig 2C, D) may have to be moved to the supplementary materials. There are also a few panels/analyses that do not help the storyline directly.

Answer: Thank you for your thorough and constructive feedback. We have reorganized the manuscript, by taking out the cross-tissue section and moving up the microbiome section (high-level analysis) to the beginning of the story. We have also removed direct comparisons between diet and genetics, as well as the schematics to avoid confusion. We agree with the reviewer in that our study provides less of a specific “storyline”, but feel that the cumulative datasets, analysis and validation will be informative to analyses of interaction between genetics and NAFLD. Panels 2C-D have also been moved to the supplement

Major comments:

1. The authors described that they used 268 male mice in the Methods section, but listed 559 mice in Fig. 2A. How many mice were used exactly?

Answer: We apologize for the incorrect values. The correct number of animals are actually 279. This was due to the fact that 559 mice have been assayed for different molecular and clinical traits. Lipidomics was measured in 279 mice (102 strains, n = 1-4 mice per strain) and we have corrected texts/legends accordingly. For the measures of dietary impact (Fig 1C), we used n = 3 mice per strain and diet group when we compared the HF/HS vs normal chow. This information has also been added to the text and the legends.

2. How did the authors compute the lipid changes with genetics in Fig. 1C? The method part describing the statistics is not very clearly written.

Answer: Initially, we performed simple t-tests on abundances between genotypes (BL6 vs DBA or DBA vs C3H). Since, we additionally ran variance tests to evaluate interactions; however, all 3 reviewers felt that these comparisons do not contribute to the overall story. Further, we agree that the low numbers and specific context of comparisons could be misleading. We have decided to remove this analysis from the study

3. In general, the method part does not seem complete. The authors need to explain their methodology with more details.

Answer: By a mistake the microbiome section of the method was not included in the first draft. It has now been added to the manuscript (see answer #2 to reviewer 2). We have also expanded the methods to include specifics on lipid module construction, other statistical approaches and experimental parameters.

4. There are some computational methods to quantify the proportion of variance explained by a given factor like diet. They can use those methods instead of what they did for Fig 1C.

Answer: This is a valid point. As mentioned above, we tried several approaches before ultimately deciding to remove the analysis from the study.

5. The schemes of the figure (for example Fig. 2A, 3C) are very confusing.

Answer: We have removed all of the schematics, with the exception of Figure 2A, which provides an illustration of the study design and datasets used.

6. The authors focus their attention on the positive correlation between hepatic TAG levels and body weight or plasma insulin (Fig. 3C) but this is not the strongest correlation result. Why this choice?

Answer: We agree that other lipid species (modules) deserves more attention as well. We chose to focus on the TAGs because later in the manuscript we are describing how Map2k6 affects hepatic TAG. We have now added more information about other lipid species to the paragraph: “Lipid species clustered into 12 discrete modules, some were predominantly a single class, whereas others included lipids from multiple classes (**Fig. 3C**). For example, a majority of the TAGs (36/47) and PCs (9/22) clustered into single modules (turquoise and magenta). The turquoise and magenta modules both showed strong positive correlations with body weight and plasma insulin concentration (**Fig. 3C**). All the CLs (13/13) clustered into a single module (blue) which showed negative associations with liver cholesterol, and plasma HDL, TAG and glucose (**Fig. 3C**). Other modules were more diverse in their membership, but still showed strong correlations with phenotypic traits. For example, the purple module contained lipid species from 7 different classes (**Supplemental Fig. 2**). When combined, these lipids showed significant correlations with body weight as well as plasma insulin and HDL (**Fig. 3C**).”
Page 5, line 173

7. The authors also stated that the members of each lipid category are largely correlated, if so, how did they consider potential confounding factors for the follow-up analysis using correlation?

Answer: Because the members of each lipid category are largely correlated, we have tried not to focus too much on individual lipid species in the manuscript. But we acknowledge that it can be a limitation of the study. We have added the discussion: “Because the members of each lipid category are largely correlated, it is important to interpret single correlations between lipid species and traits with caution.”
Page 10, line 403

8. Since the authors have generated a new lipidomics dataset in the context of this manuscript, why did they use the previous GWAS significance threshold from Bennett et al, 2010?

Answer: We have used this GWAS threshold for dozens of studies within the HMDP. Like that of human studies, it is mostly adjusted for the number of SNPs assayed and their linkage structure. That being said, we appreciate that the overall structure of liver lipids and their associations might be different for new datasets. Therefore, we have added the following to the discussion:

“It is worth noting that we used the same GWA significance threshold as previous HMDP studies mentioned above. While this threshold has been robust across HMDP studies, there are many considerations for interpreting GWAS results. For example distribution of traits, population structure and allele frequencies within a population can influence results of GWAS. For these and other reasons, it is key to integrate GWAS results with other analyses (ex. GWAS of multiple biological layers or correlation structure) and experimentation to gain confidence in underlying mechanisms. “
Page 9 line 325

In addition to this discussion, we have also mentioned several approaches to interpreting GWAS loci, such as mediation:

“In selecting candidate genes to test, there are several important considerations. For example, most genes in linkage disequilibrium will be correlated with each other, making it difficult to infer a single causal candidate. Causal inference tests, such as cis-expression correlation or mediation analyses, can help to address these constraints. “ Page 9, line 343

9. If the association between *pex16* and several LPCs was demonstrated through a genetic marker, why did they perform correlation analysis in Fig 4A? Alternatively, they could apply a causal inference method.

Answer: It is reasonable to assume that if an association is quite strong between a series of SNPs to both a gene and trait, that the gene and trait will be correlated. This does not appear to always be the case, possibly due to the complexity of integrating eQTL data with other associations (eg liver lipids). In this study, only 42% of particularly striking associations found in cis-eQTLs (+/- 1mB from a gene and p value < $1e-8$) which co-map to lipids actually show correlation between the gene and lipid. While correlation analysis is far-from perfect, we feel as though it further adds confidence on top of the association. For this reason, we have included in Fig 4, 5 and 6.

The suggestion of application of a causal method is valid. We have added cis-expression analysis to the revised supplement to allow readers to query a subtype of further inference methods.

10. The authors picked *Map2k6* as the candidate regulator for TAGs in Fig. 5, how was this gene prioritized over the nearby *Abca* genes (potential transporters of lipids, ref: PMID: 16586097)?

Answer: See answer #4 to reviewer 2. Shortly, *Map2k6* was a strong candidate, and there were unfortunately no probes for *Abca5* and *Abca6* on our microarray platform.

11. What is the GWAS result for PC(C38:3) in Fig. 6? Can the authors provide the Manhattan and regional plots? Also, it is not as the authors claimed that *Ifi203* is the most proximal gene of the peak SNP (the peak SNP rs31614030 is located at chr1 - 173,308,737, while the *Ifi203* gene is located at chr1 - 173,920,400). There are quite a few genes between the peak SNP and *Ifi203* gene, for example, *Aim2* and a cluster of other *Ifi* genes. Why did the authors pick *Ifi203* over the others? Is there any connection/co-regulation between *Ifi203* and other *Ifi* genes?

Answer: We have changed the text to indicate the fact that the gene is not the closest to the peak SNP, rather the strongest in the locus:

“We identified a locus (peak SNP at rs31614030) significantly associated with expression of a proximal gene, Interferon-activable protein 203, *Ifi203*, hepatic PC(C38:3) levels, and plasma insulin concentrations (Fig. 6A-D). In addition, a strong correlation was observed between the lipid levels, gene expression, and phenotypic traits (Fig. 6E-G). We note that this was the only examples where a loci comapped to a gene in *cis*, multiple lipid species and a clinical trait; where significant correlation was also observed between these measures. While other genes (including interfereon-activatable

family members) within the same locus showed significant associations to the peak SNP, albeit not as significant, *Ifi203* was the only one which also correlated in directions consistent with the genetic effects. “ Page 7, Line 267

The Manhattan plot for *Ifi203* expression has also been provided in the main figure (Fig 6A). While *Ifi203* did not show obvious correlation with its family members, we believe that this reviewer raises an important point. Specifically, most genes within LD will be correlated and therefore, difficult to pick which could be causal. Therefore, we have added the following to our discussion:

“In selecting candidate genes to test, there are several important considerations. For example, most genes in linkage disequilibrium will be correlated with each other and therefore, difficult to infer a single causal candidate. Causal inference tests, such as cis-expression or mediation analyses can help to address these constraints“ Page 9, line 343

12. The validation experiments are weak. First of all, the timeline of the experiments, as well as the AAV approach used to assess the biological relevance of *Map2k6* (Fig. 5) and *Ifi203* (Fig. 6) are different. The hepatic overexpression of *Map2k6* does not affect plasma lipid levels: how do the authors explain the effect seen on body fat and plasma glucose/insulin levels? Along the same lines, the adenovirus system used to reduce *Ifi203* expression is not liver-specific and the metabolic effects observed are not significant (Fig. 6L and 6M).

Answer: Both AAVs and adenoviruses are reasonable ways of perturbing hepatic lipids. Whereas AAVs are used for long-term effect, the effect of adenoviruses last only up until 2 weeks. Because we wanted to be certain to observe an effect of *Map2k6* on hepatic steatosis (hepatic TAG concentration), we chose to both feed the mice a HF/HS diet for 8 weeks and to use AAV to get a long-term over-expression of *Map2k6* for 8 weeks. For *Ifi203*, we hoped that the levels of PC would change within nine days and we thus used adenovirus.

The mice injected with AAV to over-express *Map2k6* were also leaner. A leaner phenotype will often result in lower plasma insulin and glucose levels. However, we have not studied in detail why over-expression of *Map2k6* results in these phenotypes. We have now added this sentence to the discussion about *Map2k6*: “It is likely that the effects that we observed of *Map2k6* on hepatic TAGs and plasma insulin and glucose are at least partly explained by reduced adiposity.” Page 10, line 372

Although the adenovirus systems used to reduce *Ifi203* expression is not liver-specific, adenoviruses are generally mostly taken up by the liver. We were happy to observe that we could validate the predicted effect of *Ifi203* on liver PC. The reason that we did not observe significant effect of *Ifi203* knock-down can be explained by time or degree of knock-down of *Ifi203* by the adenovirus. We have now discussed accordingly: “Genetic variation affecting the level of *Ifi203* expression was predicted to correlate positively with both hepatic PC and plasma insulin levels. This relationship was validated experimentally for hepatic PC, where reduction of liver *Ifi203* expression via adenovirus led to increased total PC levels. The fact that we only observed a trend towards increased plasma insulin concentration after reduction of liver *Ifi203* expression, might be explained by limited time of gene knock-down by adenoviral treatment or degree of *Ifi203* knockdown.” Page 9, line 347

13. Fig. 5F why is there no endogenous *Map2k6* detected in the AAV-GFP mice? Is it because the levels seen in the overexpressed group are so high that the endogenous one becomes proportionally undetectable? If so, this really questions the validity of an experiment where the gene of interest is overexpressed at supra-physiological levels...The authors might want to lower the concentration of virus used.

Answer: We were pleased to observe the predicted effect of *Map2k6* on hepatic TAG in AAV-treated mice. However, we agree that the *Map2k6* was substantially over-expressed. We have now provided a blot where you can observe *Map2k6* in wild-type mice. We have also discussed the finding in a

limitation section in the discussion: “Although we provided experimental *in vivo* evidence in mice for our murine GWAS candidate genes *Ifi203* and *Map2k6* as regulators of accumulation of specific classes of liver lipids, we did not both over-express and knock down the genes in mice. We also did not test if we could get a dose dependent effect on hepatic lipids with different concentrations of adenoviruses and AAVs. For example, we cannot exclude that less over-expression of *Map2k6* would result in a different phenotype.” Page 10, line 404

14. The analysis of the gut microbiome and its link with the liver lipidome is very superficial and does not add much to this manuscript. The same is true for the transcriptome analysis of the other metabolic organs (Fig. 7C and 7D).

Answer: We have decided to take out the cross-tissue section (Fig. 7D) and move up the microbiome section (Fig 2). We have also extended the result and discussion section about the gut microbiome to include module correlation. We agree that, while results are merely correlative, analysis of correlation structure can be insightful.

15. In Fig. 7D, the authors used a method they previously described to study the endocrine circuits using gene expression data across tissues. However, the values in Fig. 7D are very different from the ones in their original paper, where the final scores were at most 2~5 (e.g. Fig. 2 in Seldin 2018). Why are these scores so different? This method uses gene expression and not secretomics data. How valid are these results?

Answer: We have decided to take out the cross-tissue section (Fig. 7D). We still feel that the analysis is valid and the discrepancy in raw values are due to the large difference in number of lipids being measured (as opposed to global transcriptomics in the original paper); however, distracts from the overall message of the study

Minor comments

1. Text in many figures is squeezed.

Answer: We have fixed to text in the different figures

2. There are many typos in the text and figures.

Answer: We have gone through the manuscript to correct typos.

3. Module color labels in Fig3C and FigS2 are different.

Answer: We apologize for the oversight and have corrected the labels. While the order is different (due to hierarchical clustering based on either trait or microbiome relationships), the modules are the same.

4. The sample size is not mentioned for many figures, for example, Fig5G-K.

Answer: We have now included sample size in figure 1, 5 and 6.

5. Fig 4D is missing.

Answer: We have now fixed the nomenclatures for figure 4 in the result section.

Thank you for sending us your revised manuscript. We have now heard back from the three reviewers who agreed to evaluate your manuscript. As you will see below, while Reviewers #1 and #2 are satisfied with the revision, Reviewer #3 still raised a couple of concerns with regards to the potential bias in the lipidomic analysis, the validation for Lfi203 and the association between the peak SNP to Lfi203. During our "pre-decision cross-commenting" process (in which the reviewers are given the chance to make additional comments, including on each other's reports), Reviewer #2 thinks that these concerns raised by Reviewer #3 could be clarified and the potential limitations in this regard need to be discussed. We would therefore ask you to clarify and discuss these issues in the revised manuscript. The other minor issues raised by Reviewer #3 need to be addressed.

On a more editorial level, please address the following issues.

REFEREE REPORTS

Reviewer #1:

The authors have adequately addressed this reviewer's concerns and suggestions. No further comments.

Reviewer #2:

The authors did an excellent work during the revision of their paper. I recommend the publication of the paper in its current form.

Reviewer #3:

The authors have done significant efforts to improve the current study. Nevertheless, there are some concerns that remain, which are indicated below.

-The authors now specify that they performed lipidomics on 279 mice (102 strains, n= 1-4 mice per strain). In my opinion having one mouse representing a particular strain could bias the analysis. I would suggest to reduce the number of strains included in the study and keep only those with at least 2 (if the difference between the biological replicates is low) or more mice.

-This reviewer did not find the datasets EV1, EV2, EV3, EV4, EV5, EV6.

-Several panels in the figures are mislabeled. Figure 1C is missing and panel D is presented instead. Figure 2A is probably cited instead of Figure 2B at lines 137-140. Figure 3B is cited instead of Figure 3C (lines 163-165). There is no figure legend for Figure 2A.

-For the locus of TAG(48:2) QTL in Fig 5, the authors should include in the discussion that nearby Abca genes could also be possible modulators of this TAG species. They were left out from the following analysis simply because of a limitation of their microarray platform.

-Despite my initial comment on the weakness of the validation approaches used, no effort has been made to improve this point. The validation approach used for lfi203 remains weak. In addition, Figure 6A is not a Manhattan plot. The authors claimed that other genes in the locus of PC(C38:3) QTL did not associate as significant to the peak SNP as lfi203, but chose not to show those data. This reviewer would be very much interested to see the effects of this SNP on the expression of those genes in either a data table or a plot, for example a volcano plot or PheWAS Manhattan plot.

-Reg point 9: there is no supp. material to see the new analysis.

The authors have done significant efforts to improve the current study. Nevertheless, there are some concerns that remain, which are indicated below.

Answer: Thank you for the enthusiasm and helpful suggestions throughout the process

-The authors now specify that they performed lipidomics on 279 mice (102 strains, n= 1-4 mice per strain). In my opinion having one mouse representing a particular strain could bias the analysis. I would suggest to reduce the number of strains included in the study and keep only those with at least 2 (if the difference between the biological replicates is low) or more mice.

Answer: Lipidomics were performed on a single mouse liver in only two strains. Furthermore, we performed lipidomics on two, three and four livers in 23, 73 and 3 strains, respectively. We do not believe that excluding data from the two strains with lipidomics on only one mouse will affect the conclusions. While the generalized effects on association mapping and network models of 2 mice should be minimal, we have pointed out this consideration as a limitation in the discussion section:

“Further, the number of mice assayed in each strain ranged from 1-4 individuals. In particular, only one mouse was used for two strains, which could contribute to analysis bias.»

We agree with the reviewer in that a major strength of panels like the HMDP and BXD is having replicates of genetic backgrounds across measures and enabling reproducibility; however, other panels (ex. diversity outbred) implement similar analyses with only one mouse per background successfully. Our reasoning in keeping the two mice with a single individual is that this removal would require reanalysis of every observation and figure in the manuscript, likely with minimal change to any conclusions.

To further clarify this point to readers, we have added more details about the on numbers per strain in the method section

“Liver samples (about 20 mg) from 279 male mice (n = 1-4 mice per strain; n = 1 in 2 strains; n = 2 in 23 strains; n = 3 in 73 strains; n = 4 in 3 strains) were homogenized and lipids extracted by 10 vol of chloroform:methanol (v:v 2:1).”

-This reviewer did not find the datasets EV1, EV2, EV3, EV4, EV5, EV6.

Answer: All EV datasets have now been uploaded as separate files. We originally added them as a single zipped file in the previous submission and apologize for the confusion.

-Several panels in the figures are mislabeled. Figure 1C is missing and panel D is presented instead. Figure 2A is probably cited instead of Figure 2B at lines 137-140. Figure 3B is cited instead of Figure 3C (lines 163-165). There is no figure legend for Figure 2A.

Answer: We apologize for the confusion and have made sure that the text/legends match the correct figures.

-For the locus of TAG(48:2) QTL in Fig 5, the authors should include in the discussion that nearby Abca genes could also be possible modulators of this TAG species. They

were left out from the following analysis simply because of a limitation of their microarray platform.

Answer: We have now added to the discussion to point out this limitation:

“Two other candidate genes potentially regulated by this locus, *Abca5* and *Abca6*, might be transporters of lipids (Albrecht & Viturro, 2007). Unfortunately, *Abca5* and *Abca6* were not present on the liver expression arrays.”

We decided to focus on *Map2k6* because it was a very strong candidate gene. *Map2k6* expression was regulated in cis. Both *Map2k6* expression and the cis component of *Map2k6* expression correlated with TAG species. That being said, *Abca5* and *Abca6* might also affect TAG accumulation.

-Despite my initial comment on the weakness of the validation approaches used, no effort has been made to improve this point. The validation approach used for *Ifi203* remains weak. In addition, Figure 6A is not a Manhattan plot. The authors claimed that other genes in the locus of PC(C38:3) QTL did not associate as significant to the peak SNP as *Ifi203*, but chose not to show those data. This reviewer would be very much interested to see the effects of this SNP on the expression of those genes in either a data table or a plot, for example a volcano plot or PheWAS Manhattan plot.

Answer: Similar to findings for the *Map2k6* locus, the reviewer raises a valid point. Further, while we found it encouraging that two weeks of *Ifi203* reduction altered the levels of lipid and genes corresponding to the hypothesized pathway, other phenotypic effects (ex Insulin) were modest. For the peak SNP of PC and Insulin, we have provided a visual of genome track view (revised Fig EV6A) and the corresponding cis-eQTL data for each gene detected in our arrays (revised Fig EV6B). This has been mentioned in the text:

«While other genes (including interferon-activable family members) within the same locus showed strong associations to the peak SNP, albeit not as significant, *Ifi203* was the only one which also correlated in directions consistent with the genetic effects. The surrounding genome view of the locus and pvalue of all genes detected on our arrays are provided in **Fig. EV6.**»

Since several other genes in this region might present interesting candidates for experimentation, but were not detected in our arrays, we have also elaborated in the discussion:

“Given that many *Ifi* genes are also locally regulated by SNPs in this locus, and that other candidates were not available in expression arrays, we cannot exclude that other candidates than *Ifi203* also affects hepatic PC levels.»

-Reg point 9: there is no supp. material to see the new analysis.

Answer: All EV datasets have now been uploaded as separate files. We originally added them as a single zipped file in the previous submission and apologize for the confusion.

Thank you again for sending us your revised manuscript. We are now satisfied with the modifications made and I am pleased to inform you that your paper has been accepted for publication.

Corresponding Author Name: Marcus Seldin and Aldons Jake Lusis

Manuscript Number: MSB-20-9684